# Dual-branch Graph Domain Adaptation for Cross-scenario Multi-modal Emotion Recognition

## Abstract

Multimodal Emotion Recognition in Conversations (MERC) aims to predict speakers' emotional states in multi-turn dialogues through text, audio, and visual cues. In real-world settings, conversation scenarios differ significantly in speakers, topics, styles, and noise levels. Existing MERC methods generally neglect these cross-scenario variations, limiting their ability to transfer models trained on a source domain to unseen target domains. To address this issue, we propose a Dual-branch Graph Domain Adaptation framework (DGDA) for multimodal emotion recognition under cross-scenario conditions. We first construct an emotion interaction graph to characterize complex emotional dependencies among utterances. A dual-branch encoder, consisting of a hypergraph neural network (HGNN) and a path neural network (PathNN), is then designed to explicitly model multivariate relationships and implicitly capture global dependencies. To enable out-of-domain generalization, a domain adversarial discriminator is introduced to learn invariant representations across domains. Furthermore, a regularization loss is incorporated to suppress the negative influence of noisy labels. To the best of our knowledge, DGDA is the first MERC framework that jointly addresses domain shift and label noise. Theoretical analysis provides tighter generalization bounds, and extensive experiments on IEMOCAP and MELD demonstrate that DGDA consistently outperforms strong baselines and better adapts to cross-scenario conversations. Our anonymous code is available at https://anonymous.4open.science/r/DGDA-Net-1A58.

## 1 Introduction

Multimodal emotion recognition in conversations (MERC) Liu et al. (2025); Wang et al. (2025); Ai et al. (2025); Yang et al. (2024a); Zhang et al. (2023) aims to predict the emotional state of participants in a multi-round conversation through multimodal information (e.g., text, audio, and video) and has broad application prospects in dialogue generation Ghosh et al. (2017); Tu et al. (2024); Zhang et al. (2024a); Tellamekala et al. (2023); Wen et al. (2024), social media analysis Khare & Bajaj (2020); Peng et al. (2024); Xu et al. (2025); Huang et al. (2020), intelligent systems like smart homes and chatbots Young et al. (2018); Lu et al. (2025); Kang & Cho (2025); Li et al. (2025).

In the MERC task, researchers mainly focus on learning the emotional feature representation of in-domain data. As shown in Fig. 1 (a), a well-crafted encoder-classifier architecture is used to achieve multimodal emotion recognition without considering out-of-domain distribution differences. The mainstream MERC method mainly uses Transformers Hazmoune & Bougamouza (2024); Lian et al. (2021); Ma et al. (2023) and graph neural networks (GNN) Yang et al. (2024b); Chen et al. (2023) as the encoder to model contextual dependency information and speaker dependency. Although existing methods have achieved relatively good emotion recognition results, they ignore the impact of cross-domain distribution differences on emotion recognition performance. In other words, there may be significant differences in language styles, emotional expressions, and contextual environments in different domains, and these differences may lead to the limited generalization capabilities of the model. Furthermore, if some samples in the dataset are incorrectly labeled, e.g., the emotion of anger is incorrectly labeled as neutral, then the model may learn these wrong patterns during training. As a result, the model may misclassify the angry emotion as neutral in practical applications, thus affecting the accuracy and reliability of emotion recognition Lian et al. (2023); Wagner et al. (2023).

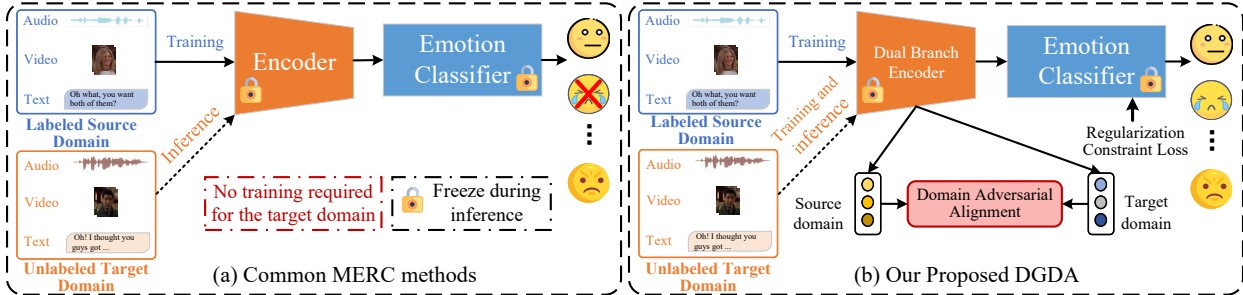

Figure 1: (a) Common MERC methods. A well-crafted encoder architecture is used to achieve multimodal emotion recognition without considering out-of-domain distribution differences. (b) Our Proposed Dual-branch Graph Domain Adaptation (DGDA) method. DGDA exploits a dual-branch encoder to explicitly and implicitly extract multimodal features, and constructs a domain adversarial alignment strategy and regularization loss to achieve out-of-domain distribution data generalization and resistance to noise label interference.

To address the above problems, we propose a Dual-branch Graph Domain Adaptation (DGDA) for multimodal emotion recognition in cross-scenario conversations, as shown in Fig. 1 (b). Specifically, to capture the discriminative features of emotion in multimodal utterances, we first construct an emotion interaction graph to model the complex emotional dependencies between utterances. Then, we design a hypergraph aggregation and path aggregation dual-branch graph encoder to explicitly and implicitly capture the dynamic changes in emotion between utterances and explore multivariate relationships, respectively. To address the problem of out-of-domain distribution differences, we introduce a domain adversarial classifier to improve the representation ability of invariant features in the source domain. In addition, we construct a regularization loss to prevent the model from memorizing noise and improve the model's ability to resist interference from noisy labels in source domains. Extensive experiments and evaluations demonstrate DGDA's superiority.

The main contributions of this paper are summarized as follows:

- To the best of our knowledge, we make the first attempt to simultaneously mitigate domain shift and noisy label interference problems in MERC scenarios, thereby enhancing usability in real-world scenarios.

- We improve the expressiveness of domain-invariant features of the original graph by introducing a domain adversarial classifier and solving the problem of out-of-domain distribution differences.

- We added a regularization constraint loss on the basis of the cross-entropy loss term to effectively suppress the model's over-learning of noisy labels and encourage the model to pay more attention to the real signals in the data.

- We provide theoretical proof to ensure that the designed DGDA is more precisely tailored for cross-scenario conversations. Extensive experiments conducted on the IEMOCAP and MELD datasets showed that DGDA is significantly better than existing baseline methods.

## 2 Related work

### 2.1 Multimodal Emotion Recognition in Conversations

Multimodal emotion recognition in conversations (MERC) has emerged as a key research area in artificial intelligence, especially at the intersection of natural language processing (NLP), computer vision (CV), and speech processing Li et al. (2024b); Liu et al. (2024a); Tao et al. (2025); Qin et al. (2025). Its objective is to infer human emotional states by jointly analyzing textual content, acoustic cues, and visual expressions Sun et al. (2024); Chen et al. (2024); Guo et al. (2025); Tang et al. (2025). RNN-based MERC methods

primarily focus on extracting contextual semantic information by modeling sequential dependencies within multimodal inputs through recurrent memory units Majumder et al. (2019); Huddar et al. (2021); Ho et al. (2020). Transformer-based methods leverage self-attention and multi-head attention mechanisms, often combined with pretrained language models, to capture long-range dependencies in conversations and achieve more effective multimodal fusion Zhao et al. (2023); Ma et al. (2023). Meanwhile, GCN-based approaches utilize the structural flexibility of graph convolutional networks to model inter-utterance relations, multimodal associations, and latent interaction patterns within dialogues Ren et al. (2021); Yuan et al. (2023); Ai et al. (2024). *Despite their effectiveness within individual datasets, these methods generally overlook the challenges of cross-scenario multimodal emotion recognition and exhibit limited generalization when applied to out-of-domain conversational distributions.* Existing models often rely heavily on dataset-specific characteristics and struggle to maintain stable performance when domain shifts arise, such as variations in conversation styles, recording conditions, speaker demographics, or modality quality. This vulnerability leads to degraded robustness and restricts the deployment of MERC systems in real-world, heterogeneous environments. *In contrast, as illustrated in Fig. 2, our proposed DGDA framework explicitly addresses this limitation by introducing a domain adversarial classifier. This component encourages the model to learn domain-invariant feature representations through an adversarial optimization process, thereby mitigating domain discrepancies between the source and target distributions. By enhancing the extraction of shared, stable, and transferable multimodal features, DGDA significantly improves the model's capability to generalize to out-of-domain conversational datasets and ensures more robust emotional understanding across diverse real-world scenarios.*

## 2.2 Graph Domain Adaption

Graph domain adaptation is a core issue in graph transfer learning Qiu et al. (2020); Sun et al. (2022); Liu et al. (2024b); Zhang et al. (2025); Shou et al. (2024) and has received increasing attention in recent years, particularly in fields such as social networks and molecular biology You et al. (2023); Chen et al. (2025); Zhang et al. (2024c). Early studies mainly focused on transferring knowledge from a well-labeled source graph to an unlabeled target graph, aiming to learn discriminative representations for target graph nodes through label supervision from the source domain Wu et al. (2020); Jin et al. (2024); Dan et al. (2024). These methods generally rely on propagating information along graph topology so that the target graph can inherit semantic cues and structural priors from the source graph. More recent research has further extended this paradigm to the graph-level setting, where multiple labeled source graphs must guide an unlabeled target graph Yang et al. (2020); Hu et al. (2024). In this scenario, the challenges go beyond simple node-level transfer; models must also handle semantic alignment, structural correspondence, and cross-graph knowledge integration at a holistic level Yin et al. (2022). Achieving such adaptation requires capturing similarities and discrepancies across heterogeneous graph distributions, reconciling different structural patterns, and transferring high-level semantic information. However, current graph-based domain adaptation methods face several fundamental limitations. Most models rely heavily on message-passing mechanisms that aggregate information from local neighborhoods. Although effective for learning localized patterns, such approaches struggle to capture high-order semantic dependencies, long-range relational structures, and global graph topology. As a result, they may fail to model complex structural variations between the source and target graphs, leading to insufficient domain alignment. In addition, existing methods typically assume that labels in the source domain are clean and reliable. In real-world scenarios, however, labeled graphs often contain noisy, ambiguous, or even contradictory annotations. Such label noise can propagate through the message-passing process, amplifying errors and degrading representation quality. The lack of explicit mechanisms to suppress noisy-label interference further limits the robustness and generalization performance of current approaches. *Therefore, more advanced graph domain adaptation methods are needed to simultaneously capture global semantic structures and provide robustness against label noise, enabling more accurate and reliable cross-graph knowledge transfer.*

# 3 METHODOLOGY

## 3.1 Task Definition

In the task of Cross-Scenario Multimodal Emotion Recognition, we aim to build robust emotion recognition models that can generalize across diverse domains or scenarios, such as different datasets, environments, recording conditions, or speaker groups. Formally, we assume a set of speakers $S = \{s_1, s_2, \ldots, s_M\}$ participating in emotionally rich conversations. Each conversation is composed of a sequence of utterances in chronological order, denoted as $U = \{u_1, u_2, \ldots, u_N\}$, where $N$ is the total number of utterances. Each utterance $u_i$ is associated with a speaker $s_{p_i}$, defined through a speaker mapping function $p(\cdot) : \{1, \ldots, N\} \rightarrow \{1, \ldots, M\}$. Furthermore, each utterance $u_{p_i}$ contains multimodal information, including textual modality $u_{p_i}^t$, visual modality $u_{p_i}^v$, and acoustic modality $u_{p_i}^a$. Unlike traditional emotion recognition tasks that assume training and testing data come from the same distribution, cross-scenario multimodal emotion recognition explicitly considers the domain shift between source and target scenarios. These shifts may arise due to variations in background, lighting, language usage, speaker identity, or even cultural differences. The objective is to predict the discrete emotion labels for each utterance in the target scenario, leveraging the multimodal information while ensuring robust generalization from the source to the target domain. This problem setting poses unique challenges, such as modality-specific noise, semantic gaps between scenarios, and inconsistent emotion distributions. Therefore, effective cross-scenario multimodal emotion recognition models must learn domain-invariant yet emotion-discriminative representations across modalities and scenarios.

## 3.2 Multimodal Feature Extraction

We extract unimodal features at the utterance level as follows. Following our previous work Ma et al. (2023), we introduce the RoBERTa Large model Kim & Vossen (2021) for text feature extraction. The dimension of the final text feature representation is 1024. We use openSMILE Eyben et al. (2010) to extract acoustic features. After feature extraction using openSMILE, we perform dimensionality reduction on the acoustic features through fully connected (FC) layers, reducing the feature dimension to 1582 for the IEMOCAP dataset and 300 for the MELD dataset. We use the DenseNet model Huang et al. (2017) pre-trained on the Facial Expression Recognition Plus dataset for visual feature extraction. In the process of visual feature extraction, we use the output dimension of DenseNet as 342.

## 3.3 Modal Feature Encoding

For the MERC, the original dimensionality spaces of text, visual, and acoustic modalities are usually significantly different, which makes them not directly usable for graph construction or fusion. To address this problem, we design a shallow feature extractor that contains three independent encoders to map them into the same dimensional space. For text modality, we use a bidirectional gated recurrent unit (Bi-GRU) Poria et al. (2017) to capture the bidirectional dependencies of context. However, through empirical observation Chen et al. (2023), we find that using recurrent neural network modules to encode visual and acoustic modalities does not bring performance improvements. Therefore, we use a simple and efficient linear layer to convert them to the same dimensional space as the text modality. After feature extraction, we then input it into the subsequent emotion reasoning network.

## 3.4 Dual Branch Encoder

**Hypergraph neural network (HGNN) branch.** Traditional graph neural networks (GNNs) usually represent graph structures through nodes and edges, while hypergraph neural networks (HGNNs) further extend this framework by allowing a hyperedge to connect multiple nodes, thereby effectively modeling complex relationships and high-order dependencies. Specifically, given a sequence of utterances containing $N$ dialogue turns, we construct a hypergraph $H = (V, E, H)$, where $V$ is a set of nodes, $E$ is a set of hyperedges, and $H$ is an incidence matrix. Each node $v \in V$ corresponds to a unimodal utterance. Each hyperedge $e \in E$ encodes high-order dependencies between multimodal data and is assigned a weight $w(e)$. Each hyperedge $e \in E$ and each node $v \in V$ associated with $e$ is also assigned a weight $w_e(v)$. Incidence

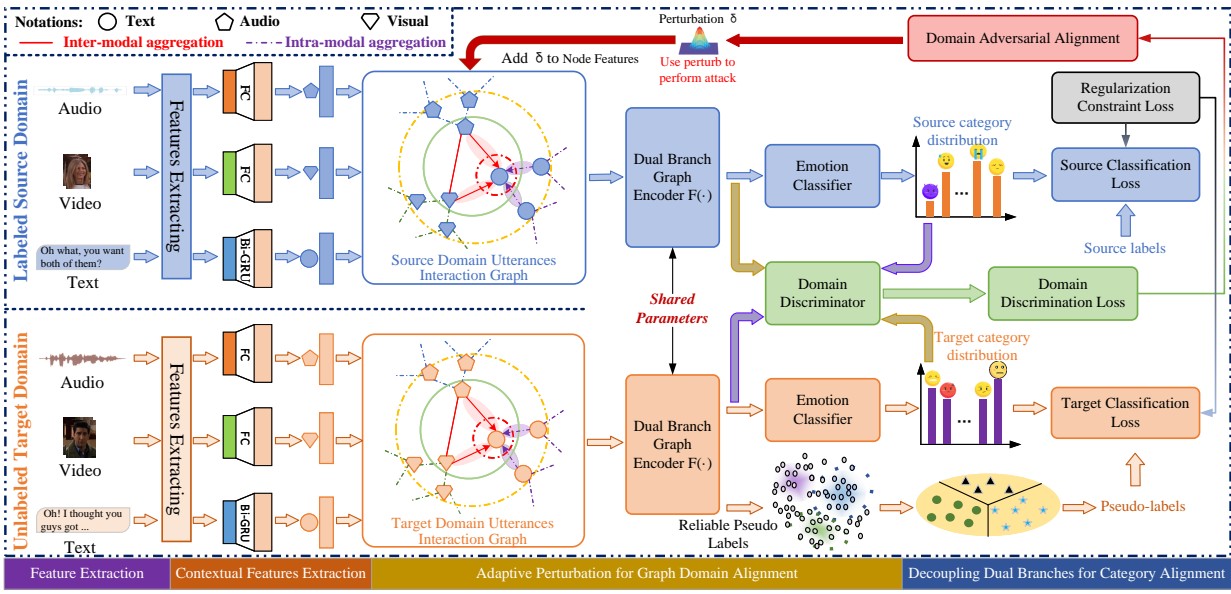

Figure 2: An overview of the proposed DGDA framework. The model operates on a labeled source domain and an unlabeled target domain. In both domains, audio, visual, and text features are first extracted and used to construct utterance-level interaction graphs. A dual-branch graph encoder encodes these graphs. For domain alignment, the source domain is adaptively perturbed by a learned noise $\delta$, while a domain discriminator promotes feature invariance across domains. Meanwhile, category-level alignment is enforced by coupling the dual-branch outputs. The final emotion classifier is trained using source labels and pseudo-labeled target samples.

matrix $H \in \mathbb{R}^{|V| \times |E|}$ represents the relationship between nodes and hyperedges. If node $v$ is associated with hyperedge $e$, then $H_{ve} = 1$, otherwise $H_{ve} = 0$. Following the paradigm of previous work Chen et al. (2023), we adopted the node-hyperedge information interaction mechanism to achieve iterative update and fusion of features by alternately performing node convolution and hyperedge convolution. Mathematically,

$$\mathbf{H}^{(l+1)} = \sigma \left( \mathbf{D}^{-1} \mathbf{H} \mathbf{W}_e \mathbf{B}^{-1} \hat{\mathbf{H}}^\top \mathbf{H}^{(l)} \right), \tag{1}$$

where $\mathbf{H}^{(l)} = \{v_{i,(l)}^x | i \in [1, N], x \in \{t, a, v\}\} \in \mathbb{R}^{|\mathbf{H}| \times D}$ is the input at layer $l$, $v_i^t, v_i^a, v_i^v$ is the textual modality, acoustic modality and visual modality, respectively, $\mathbf{W}_e = \text{diag}(w(e_1), ..., w(e_{|E|}))$ is the hyperedge weight matrix, $\mathbf{B}$ and $\mathbf{D}$ are the hyperedge degree and node degree matrix, respectively.

**Path neural networks (PathNN) branch**. To effectively capture the global dependencies between nodes, we introduced PathNN Michel et al. (2023) into the model to make up for the lack of modeling capabilities of long-distance node relationships based on neighbor aggregation methods when dealing with complex graph structures. We first extract the feature representations of each node in the path based on the predefined paths in the graph, and then fuse the information of the nodes in the path into path representations through path aggregation operations. Finally, these path representations will be reversely injected into the start and end nodes associated with the path to achieve feedback of path information and enhancement of node representation. Suppose there is a set of paths $\mathcal{P}$, each path is represented as $p = (v_1, v_2, ..., v_l)$, where $v_i$ is the $i$-th node in the path and $l$ is the path length. To further improve the expressiveness of node feature aggregation within a path and dynamically capture the importance of different nodes in path representation, we introduced an aggregation method based on the attention mechanism. We first calculate the attention score $\alpha_{vi}$ for each node in the path, defined as follows:

$$\alpha_{v_i} = \frac{\exp \left( \text{LeakyReLU} \left( \mathbf{a}^\top [\mathbf{W} \mathbf{h}_{v_i} \| \mathbf{c}_p] \right) \right)}{\sum_{j=1}^{l} \exp \left( \text{LeakyReLU} \left( \mathbf{a}^\top [\mathbf{W} \mathbf{h}_{v_j} \| \mathbf{c}_p] \right) \right)}, \tag{2}$$

where $\mathbf{W}$ is a learnable matrix, $c_p$ is the path global context vector, $\mathbf{a}$ is a learnable attention weight vector, and $[\cdot \ || \ \cdot]$ represents a vector concatenation operation. Next, the path representation $h_p$ is obtained by weighted summing of the node features in the path according to the attention weights:

$$\mathbf{h}_p = \sum_{i=1}^{l} \alpha_{v_i} \cdot \mathbf{W} \mathbf{h}_{v_i}. \tag{3}$$

Finally, the path information is passed back to the path-related nodes and updated as follows:

$$\mathbf{h}_{v_i}^{(new)} = \mathbf{h}_{v_i}^{(old)} + \sum_{p \in \mathcal{P}(v_i)} \frac{1}{|\mathcal{P}(v_i)|} \mathbf{h}_p, \tag{4}$$

where $\mathcal{P}(v_i)$ is the set of paths associated with node $v_i$.

### 3.5 Adversarial Alignment for Domain Adaptation

To alleviate the distribution difference between the source domain and the target domain in the multimodal graph semantic space and promote the consistency of the feature space, we introduced the adaptive perturbation and adversarial alignment mechanisms in both the HGNN branch and the PathNN branch. The core idea is to enhance the robustness of the model to domain changes by introducing learnable perturbations in the feature space, and dynamically optimizing the distribution of source domain and target domain features with the help of adversarial training to align them in the feature space. Specifically, for the original node feature $\mathbf{H}$ extracted by the HGNN branch and the PathNN branch, we first generate a perturbation vector for it through the adaptive perturbation generator and inject it into the original feature to obtain the perturbed feature representation:

$$
\begin{aligned}
\widetilde{\mathbf{H}}^{HGNN} &= \mathbf{H}^{HGNN} + \delta^{HGNN} \cdot \mathbf{M}(\mathbf{H}^{HGNN}), \\
\widetilde{\mathbf{H}}^{PathNN} &= \mathbf{H}^{PathNN} + \delta^{PathNN} \cdot \mathbf{M}(\mathbf{H}^{PathNN}),
\end{aligned}
\tag{5}
$$

where $\delta$ is the perturbation intensity coefficient and $\mathbf{M}(\cdot)$ is the perturbation generation network, usually MLP. Next, we designed a domain discriminator to distinguish whether the features come from the source domain $\mathbf{H}_{src}$ or the target domain $\mathbf{H}_{tgt}$ as follows:

$$\mathcal{L}_D = -\mathbb{E}_{\widetilde{\mathbf{H}}_{src}}[\log D(\widetilde{\mathbf{H}}_{src})] - \mathbb{E}_{\widetilde{\mathbf{H}}_{tgt}}[\log(1 - D(\widetilde{\mathbf{H}}_{tgt}))]. \tag{6}$$

The feature extractor (i.e., HGNN and PathNN branch) optimization objectives are as follows:

$$\mathcal{L}_{adv} = -\mathbb{E}_{\widetilde{\mathbf{H}}_{tgt}}[\log D(\widetilde{\mathbf{H}}_{tgt})]. \tag{7}$$

During the training process, the domain discriminator and feature extractor are optimized alternately. The former attempts to correctly distinguish the source/target domain features, while the latter is continuously optimized by introducing perturbations and adversarial training so that the distributions of the two gradually converge.

### 3.6 Branch Coupling for High-confidence Pseudo-label Generation

In the cross-session multimodal emotion recognition task, the source domain data has reliable emotion labels, while the target domain has the problem of missing labels. Directly assigning pseudo labels to target samples for training often introduces noise due to low-confidence labels, affecting model performance. To this end, we propose branch coupling to generate high-confidence pseudo labels, aiming to make full use of the complementary features of the HGNN $p_\theta$ and the PathNN $q_\phi$ branch in graph semantic modeling and jointly optimize the quality of pseudo labels. We maximize the evidence lower bound (ELBO) of the

log-likelihood with the source and target labels $y_s$ and $y_t$:

$$\log p_\theta(y^s|G^s, G^t) = \log \int p_\theta(y^s, y^t|G^s, G^t) \, dy^t = \log \int \frac{p_\theta(y^s, y^t|G^s, G^t)}{q_\phi(y^t|G^t)} q_\phi(y^t|G^t) \, dy^t$$

$$\geq \int q_\phi(y^t|G^t) \log \frac{p_\theta(y^s, y^t|G^s, G^t)}{q_\phi(y^t|G^t)} \, dy^t \tag{8}$$

$$\geq \mathbb{E}_{q_\phi(y^t|G^t)} \left[ \log p_\theta(y^s, y^t|G^s, G^t) - \log q_\phi(y^t|G^t) \right].$$

The traditional ELBO optimization strategy is mainly used to learn the approximate distribution $q$ of the target distribution $p$. However, the HGNN branch and the PathNN branch can model the potential label distribution of the samples respectively, one of which acts as the target distribution $p$, providing a relatively stable teacher signal, and the other as the approximate distribution $q$, which is as close to the target distribution as possible by maximizing the ELBO. To be specific, the ELBO in Eq. 8 can be equivalently written as:

$$\mathbb{E}_{q_\phi(y^t|G^t)} \left[ \log p_\theta(y^t|G^s, G^t, y^s) p_\theta(y^s|G^s) - \log q_\phi(y^t|G^t) \right]$$

$$= \mathbb{E}_{q_\phi(y^t|G^t)} \left[ \log \frac{p_\theta(y^t|G^s, G^t, y^s)}{q_\phi(y^t|G^t)} \right] + \mathbb{E}_{q_\phi(y^t|G^t)} [p_\theta(y^s|G^s)] \tag{9}$$

$$= -KL(q_\phi(y^t|G^t) \parallel p_\theta(y^t|G^s, G^t, y^s)) + \mathbb{E}_{q_\phi(y^t|G^t)} [p_\theta(y^s|G^s)].$$

When optimizing the HGNN branch, we use the distribution $q_\phi$ of the PathNN branch output as the target and calculate the loss of the HGNN branch to approximate this distribution. Conversely, when optimizing the PathNN branch, we fix the distribution $p_\theta$ of the HGNN branch and use it as the optimization target of the PathNN branch. Based on the above ideas, we define the optimization loss functions of the HGNN and PathNN branches when they are updated alternately as follows:

$$\mathcal{L}_1 = \mathbb{E}_{p_\theta\left(\hat{y}_i^t|G^t\right) > \zeta} \left[ \log q_\phi\left(y_i^s, \hat{y}_i^t \mid G_i^s, G_i^t\right) - \log q_\phi\left(\hat{y}_i^t \mid G_i^t\right) \right]$$

$$- \mathbb{E}_{q_\phi(y^s, G^s)} \log p_\theta\left(y_i^s \mid G_i^s\right),$$

$$\mathcal{L}_2 = \mathbb{E}_{p_\phi\left(\hat{y}_i^t|G^t\right) > \zeta} \left[ \log q_\theta\left(y_i^s, \hat{y}_i^t \mid G_i^s, G_i^t\right) - \log q_\theta\left(\hat{y}_i^t \mid G_i^t\right) \right] \tag{10}$$

$$- \mathbb{E}_{q_\theta(y^s, G^s)} \log p_\phi\left(y_i^s \mid G_i^s\right),$$

where $\hat{y}_i^t$ is the target pseudo-label filtered by the HGNN or PathNN branch. It should be noted that we introduced a confidence threshold $\zeta$ to ensure that only those samples with higher confidence and more reliable prediction results can participate in the subsequent optimization process.

**Theorem 1.** For a deviation measure based on the label function $\max\limits_{G_1, G_2} \frac{|\hat{h}_D(G_1) - \hat{h}_D(G_2)|}{\eta(G_1, G_2)} = C_h \leq C_f C_g (D \in \{S, T\})$, let $H := \{h : G \to Y\}$ denote a set of bounded real-valued functions that map from feature space $G$ to label space $Y$, the samples closest to the source domain distribution are selected in the target domain, the empirical risk $\hat{\epsilon}_T$ in the target domain can be significantly reduced:

$$\epsilon_T(h, \hat{h}_T) \leq \frac{N_T'}{N_S + N_T'} \hat{\epsilon}_T(h, \hat{h}_T) + \frac{N_S}{N_S + N_T'} \left( \hat{\epsilon}_S(h, \hat{h}_S) + \sqrt{\frac{4d}{N_S} \log(\frac{eN_S}{d}) + \frac{1}{N_S} \log(\frac{1}{\delta})} \right)$$

$$+ \frac{N_S}{N_S + N_T'} \left( 2C_f C_g W_1(\mathbb{P}_S(G), \mathbb{P}_T(G)) + \omega \right) \tag{11}$$

$$\leq \hat{\epsilon}_S(h, \hat{h}) + \sqrt{\frac{4d}{N_S} \log(\frac{eN_S}{d}) + \frac{1}{N_S} \log(\frac{1}{\delta})} + 2C_f C_g W_1(\mathbb{P}_S(G), \mathbb{P}_T(G)) + \omega',$$

where $\omega = \min_{\|g\|_{Lip} \leq C_g, \|f\|_{Lip} \leq C_f} \{\epsilon_S(h, \hat{h}_S) + \epsilon_T(h, \hat{h}_S)\}$ and $\omega' = \min(|\epsilon_S(h, \hat{h}_S) - \epsilon_S(h, \hat{h}_T)|, |\epsilon_T(h, \hat{h}_S) - \epsilon_T(h - \hat{h}_T)|)$.

### 3.7 Model Training

In cross-scenario emotion recognition, although the source domain data has labels, a certain proportion of labeling errors are inevitable in the real-world labeling process Liu et al. (2020). Meanwhile, in the process of

generating pseudo-labels in the target domain, inaccurate pseudo-labels are also inevitable. If noisy labels are used directly for model training, it is easy to cause the model to overfit the noise samples in the two branches of HGNN and PathNN. To alleviate this problem, we propose to introduce a regularization term in the cross-entropy loss. By introducing historical prediction results, the model can suppress the high-confidence fitting of labels in the later stage of training. Mathematically:

$$\mathcal{L}_{\text{CLS}} = -\frac{1}{N} \sum_{i=1}^{N} y_i \log p_\theta(y|x_i) + \lambda \cdot \left( \frac{1}{N} \sum_{i=1}^{N} \log \left( 1 - \langle p_\theta(y|x_i), \hat{p}_i \rangle \right) \right), \tag{12}$$

where $\lambda$ is the weight coefficient and $\hat{p}_i$ is the exponential moving average (EMA) of the model's predicted probability for sample $x_i$ in the early training stage.

**Theorem 2.** *Assume the input space is $\mathcal{X} \subseteq \mathbb{R}^d$ and the label space is $\mathcal{Y} = \{1, 2, ..., K\}$. The real data distribution is $\mathcal{D}$, and the observation distribution with noisy labels is $\tilde{D}$, where the upper limit of the noise ratio is $\eta \leq 0.5$. In the early stage of training (the first $T_0$ steps), the model's prediction of clean samples satisfies $p_{\theta_i}^i(x) \approx y_i$, where $t \leq T_0$, and $(x, y)$ comes from the clean data subset $\mathcal{D}_{\text{clean}} \subset \tilde{\mathcal{D}}_2$, we then have:*

$$p_\theta(y_i|x_i) \approx \frac{\tilde{y}_i}{\tilde{y}_i + \lambda y_i(1 - p_\theta(y_i|x_i))}. \tag{13}$$

**Theorem 3.** *Assume that the model complexity is characterized by Rademacher complexity $\mathfrak{R}_n(\mathcal{F})$ Yin et al. (2019). For any $\delta > 0$, the generalization error upper bound of $\mathcal{L}_{\text{CLS}}$ satisfies with probability $1 - \delta$:*

$$\text{GenError}_{\mathcal{L}_{\text{CLS}}} \leq \text{GenError}_{\mathcal{L}_{\text{CE}}} \leq \frac{2\mathfrak{R}_n(\mathcal{F})}{\sqrt{\lambda}} + \sqrt{\frac{\log(1/\delta)}{2n}} + O\left( \frac{\eta + \epsilon}{\mu} \right). \tag{14}$$

## 4 DETAILED PROOFS

### 4.1 Proof of Theorem 1

Intuitively, by combining training samples from the target and source domains, the class distributions between the two domains can be effectively aligned. Here, we provide a theoretical analysis to support this intuition. Specifically, we prove that after introducing the class distribution alignment module, the empirical risk lower bound in the target domain can be significantly reduced compared to models without this module. Before presenting our results, we first introduce a lemma Redko et al. (2017); Shen et al. (2018); Wang et al. (2024b), which is used in our proof:

**Lemma 1.** *Let the learned discriminator $g$ be $C_g$-Lipschitz continuous, where the Lipschitz norm is defined as $||g||_{\text{Lip}} = \max\limits_{Z_1, Z_2} \frac{|g(Z_1) - g(Z_2)|}{\rho(Z_1, Z_2)}$, $\rho$ is a Euclidean distance function, and $H := \{g : Z \to Y\}$ denote a set of bounded real-valued functions defined on the input space $Z$ and mapped to the output space $Y$. Assume that the pseudo-dimension of this set of functions is $d$, i.e., $Pdim(H) = d$. A bound on the relationship between empirical risk and true risk holds for the discriminator $g \in H$ with probability at least $1 - \delta$:*

$$\epsilon_T(h, \hat{h}) \leq \hat{\epsilon}_S(h, \hat{h}) + \sqrt{\frac{4d}{N_S} \log(\frac{eN_S}{d}) + \frac{1}{N_S} \log(\frac{1}{\delta})} + 2C_f C_g W_1(\mathbb{P}_S(G), \mathbb{P}_T(G)) + \omega_T(G), \tag{15}$$

*where $\omega = \min_{||g||_{Lip} \leq C_g} \{\epsilon_S(g, \hat{g}) + \epsilon_T(g, \hat{g})\}$ denotes the model discriminative ability, and the Wasserstein distance is defined as Villani et al. (2009):*

$$W_1(\mathbb{P}, \mathbb{Q}) = \sup_{||g||_{Lip} \leq 1} \left\{ \mathbb{E}_{\mathbb{P}_S(Z)} g(Z) - \mathbb{E}_{\mathbb{P}_T(Z)} g(Z) \right\}. \tag{16}$$

Now, we present our theoretical results in the following theorem, as well as its proof.

**Theorem 1.** *For a deviation measure based on the label function $\max\limits_{G_1, G_2} \frac{|\hat{h}_D(G_1) - \hat{h}_D(G_2)|}{\eta(G_1, G_2)} = C_h \leq C_f C_g (D \in \{S, T\})$, let $H := \{h : G \to Y\}$ denote a set of bounded real-valued functions that map from feature space $G$ to label space $Y$, under the assumptions of Lemma 1 and the following assumptions:*

And we have

1. Assume a small number of pseudo-labeled independent and identically distributed samples $\{(G_n, Y_n)\}_{n=1}^{N'_T}$, where $N'_T \ll N_S$ (the number of target domain samples is much smaller than source domain samples).

2. Assume the source domain and the target domain have different label functions, satisfying $\hat{h}_S \neq \hat{h}_T$.

3. Assuming the samples closest to the source domain distribution are selected in the target domain, the empirical risk $\hat{\epsilon}_T$ in the target domain can be significantly reduced.

$$\hat{\epsilon}_T \leq \epsilon_T \leq \hat{\epsilon}_S(h, \hat{h}) + \sqrt{\frac{4d}{N_S} \log(\frac{eN_S}{d}) + \frac{1}{N_S} \log(\frac{1}{\delta})} + 2C_f C_g W_1(\mathbb{P}_S(G), \mathbb{P}_T(G)) + \omega', \qquad (17)$$

where $\omega' = \min_{||g|||_{Lip} \leq C_g, ||f||_{Lip} \leq C_f}\{\epsilon_S(h, \hat{h}) + \epsilon_T(h, \hat{h})\}$, $\hat{\epsilon}_T$ is the empirical risk on the high confidence samples, $\epsilon_S$ is the empirical risk on the target domain.

4. Assume the pseudo-dimension of this set of functions is $Pdim(H) = d$. For any function $h \in H$, with probability of at least $1 - \delta$, the following inequality holds:

$$\epsilon_T(h, \hat{h}_T) \leq \hat{\epsilon}_S(h, \hat{h}_S) + \sqrt{\frac{4d}{N_S} \log(\frac{eN_S}{d}) + \frac{1}{N_S} \log(\frac{1}{\delta})} + 2C_f C_g W_1(\mathbb{P}_S(G), \mathbb{P}_T(G)) + \omega, \qquad (18)$$

where $\omega = \min_{||g|||_{Lip} \leq C_g, ||f||_{Lip} \leq C_f}\{\epsilon_S(h, \hat{h}_S) + \epsilon_T(h, \hat{h}_S)\}$.

*Proof.* We first introduce the following inequality to be used:

$$\begin{aligned}
|\epsilon_S(h, \hat{h}_S) - \epsilon_T(h, \hat{h}_T)| &= |\epsilon_S(h, \hat{h}_S) - \epsilon_S(h, \hat{h}_T) + \epsilon_S(h, \hat{h}_T) - \epsilon_T(h, \hat{h}_T)| \\
&\leq |\epsilon_S(h, \hat{h}_S) - \epsilon_S(h, \hat{h}_T)| + |\epsilon_S(h, \hat{h}_T) - \epsilon_T(h, \hat{h}_T)| \\
&\overset{(a)}{\leq} \left|\epsilon_S(h, \hat{h}_S) - \epsilon_S(h, \hat{h}_T)\right| + 2C_f C_g W_1(\mathbb{P}_S(G), \mathbb{P}_T(G)),
\end{aligned} \qquad (19)$$

where (a) results from Shen et al. (2018) Lemma 1 with the assumption $\max(||h||_{Lip}, \max_{G_1, G_2} \frac{|\hat{h}_D(G_1) - \hat{h}_D(G_2)|}{\eta(G_1, G_2)}) \leq C_f C_g, D \in \{S, T\}$. Similarly, we obtain:

$$|\epsilon_S(h, \hat{h}_S) - \epsilon_T(h, \hat{h}_T)| \leq |\epsilon_T(h, \hat{h}_S) - \epsilon_T(h, \hat{h}_T)| + 2C_f C_g W_1(\mathbb{P}_S(G), \mathbb{P}_T(G)). \qquad (20)$$

Combining Eqs 19 and 20, we can obtain:

$$\begin{aligned}
|\epsilon_S(h, \hat{h}_S) - \epsilon_T(h, \hat{h}_T)| &\leq 2C_f C_g W_1(\mathbb{P}_S(G), \mathbb{P}_T(G)) \\
&+ \min\left(|\epsilon_S(h, \hat{h}_S) - \epsilon_S(h, \hat{h}_T)|, |\epsilon_T(h, \hat{h}_S) - \epsilon_T(h, \hat{h}_T)|\right).
\end{aligned} \qquad (21)$$

Therefore, we can derive the generalization error bound on the target domain $\epsilon_T(h, \hat{h}_T)$:

$$\begin{aligned}
\epsilon_T(h, \hat{h}_T) &\leq \epsilon_S(h, \hat{h}_S) + 2C_f C_g W_1(\mathbb{P}_S(G), \mathbb{P}_T(G)) \\
&+ \min\left(|\epsilon_S(h, \hat{h}_S) - \epsilon_S(h, \hat{h}_T)|, |\epsilon_T(h, \hat{h}_S) - \epsilon_T(h, \hat{h}_T)|\right).
\end{aligned} \qquad (22)$$

We next link the bound with the empirical risk and labeled sample size by showing, with probability at least $1 - \delta$ that:

$$
\begin{aligned}
\epsilon_T(h, \hat{h}_T) &\leq \epsilon_S(h, \hat{h}_S) + 2C_f C_g W_1\left(\mathbb{P}_S(G), \mathbb{P}_T(G)\right) \\
&\quad + \min\left(|\epsilon_S(h, \hat{h}_S) - \epsilon_S(h, \hat{h}_T)|, |\epsilon_T(h, \hat{h}_S) - \epsilon_T(h, \hat{h}_T)|\right) \\
&\leq \hat{\epsilon}_S(h, \hat{h}_S) + 2C_f C_g W_1\left(\mathbb{P}_S(G), \mathbb{P}_T(G)\right) \\
&\quad + \min\left(|\epsilon_S(h, \hat{h}_S) - \epsilon_S(h, \hat{h}_T)|, |\epsilon_T(h, \hat{h}_S) - \epsilon_T(h, \hat{h}_T)|\right) \\
&\quad + \sqrt{\frac{2d}{N_S} \log(\frac{eN_S}{d})} + \sqrt{\frac{1}{2N_S} \log(\frac{1}{\delta})},
\end{aligned}
\tag{23}
$$

and according to previous work Mohri (2018), the upper bound of the target domain generalization error $\epsilon_T(h, \hat{h}_T)$ is defined as follows:

$$
\epsilon_T(h, \hat{h}_T) \leq \hat{\epsilon}_T(h, \hat{h}_T) + \sqrt{\frac{2d}{N_T'} \log(\frac{eN_T'}{d})} + \sqrt{\frac{1}{2N_T'} \log(\frac{1}{\delta})},
\tag{24}
$$

Combining Eq. 23 and 24, we can derive:

$$
\begin{aligned}
\epsilon_T(h, \hat{h}_T) &\overset{(a)}{\leq} \frac{N_T'}{N_S + N_T'}\left(\hat{\epsilon}_T(h, \hat{h}_T) + \sqrt{\frac{2d}{N_T'} \log(\frac{eN_T'}{d})} + \sqrt{\frac{1}{2N_T'} \log(\frac{1}{\delta})}\right) \\
&\quad + \frac{N_S}{N_S + N_T'}\left(\hat{\epsilon}_S(h, \hat{h}_S) + \sqrt{\frac{2d}{N_S} \log(\frac{eN_S}{d})} + \sqrt{\frac{1}{2N_S} \log(\frac{1}{\delta})}\right) \\
&\quad + \frac{N_S}{N_S + N_T'}\left(2C_f C_g W_1\left(\mathbb{P}_S(G), \mathbb{P}_T(G)\right)\right) + \min\left(|\epsilon_S(h, \hat{h}_S) - \epsilon_S(h, \hat{h}_T)|, |\epsilon_T(h, \hat{h}_S) - \epsilon_T(h, \hat{h}_T)|\right) \\
&\overset{(b)}{\leq} \frac{N_T'}{N_S + N_T'}\left(\hat{\epsilon}_T(h, \hat{h}_T) + \sqrt{\frac{4d}{N_T'} \log(\frac{eN_T'}{d}) + \frac{1}{N_T'} \log(\frac{1}{\delta})}\right) \\
&\quad + \frac{N_S}{N_S + N_T'}\left(\hat{\epsilon}_S(h, \hat{h}_S) + \sqrt{\frac{4d}{N_S} \log(\frac{eN_S}{d}) + \frac{1}{N_S} \log(\frac{1}{\delta})}\right) \\
&\quad + \frac{N_S}{N_S + N_T'}\left(2C_f C_g W_1\left(\mathbb{P}_S(G), \mathbb{P}_T(G)\right)\right) + \min\left(|\epsilon_S(h, \hat{h}_S) - \epsilon_S(h, \hat{h}_T)|, |\epsilon_T(h, \hat{h}_S) - \epsilon_T(h, \hat{h}_T)|\right)\right) \\
&\overset{(c)}{\leq} \frac{N_T'}{N_S + N_T'}\hat{\epsilon}_T(h, \hat{h}_T) + \frac{N_S}{N_S + N_T'}\hat{\epsilon}_S(h, \hat{h}_S) + \frac{N_S}{N_S + N_T'}\left(2C_f C_g W_1\left(\mathbb{P}_S(G), \mathbb{P}_T(G)\right)\right) \\
&\quad + \min\left(|\epsilon_S(h, \hat{h}_S) - \epsilon_S(h, \hat{h}_T)|, |\epsilon_T(h, \hat{h}_S) - \epsilon_T(h, \hat{h}_T)|\right)\right) \\
&\quad + \frac{N_T'}{N_S + N_T'}\sqrt{\frac{4d}{N_T'} \log(\frac{eN_T'}{d}) + \frac{1}{N_T'} \log(\frac{1}{\delta})} + \frac{N_S}{N_S + N_T'}\sqrt{\frac{4d}{N_S} \log(\frac{eN_S}{d}) + \frac{1}{N_S} \log(\frac{1}{\delta})} \\
&= \frac{N_T'}{N_S + N_T'}\hat{\epsilon}_T(h, \hat{h}_T) + \frac{N_S}{N_S + N_T'}\hat{\epsilon}_S(h, \hat{h}_S) + \frac{N_S}{N_S + N_T'}\sqrt{\frac{4d}{N_S} \log(\frac{eN_S}{d}) + \frac{1}{N_S} \log(\frac{1}{\delta})} \\
&\quad + \frac{N_S}{N_S + N_T'}\left(2C_f C_g W_1\left(\mathbb{P}_S(G), \mathbb{P}_T(G)\right)\right) + \min\left(|\epsilon_S(h, \hat{h}_S) - \epsilon_S(h, \hat{h}_T)|, |\epsilon_T(h, \hat{h}_S) - \epsilon_T(h, \hat{h}_T)|\right)\right) \\
&= \frac{N_T'}{N_S + N_T'}\hat{\epsilon}_T(h, \hat{h}_T) + \frac{N_S}{N_S + N_T'}\left(\hat{\epsilon}_S(h, \hat{h}_S) + \sqrt{\frac{4d}{N_S} \log(\frac{eN_S}{d}) + \frac{1}{N_S} \log(\frac{1}{\delta})}\right) \\
&\quad + \frac{N_S}{N_S + N_T'}\left(2C_f C_g W_1\left(\mathbb{P}_S(G), \mathbb{P}_T(G)\right)\right) + \min\left(|\epsilon_S(h, \hat{h}_S) - \epsilon_S(h, \hat{h}_T)|, |\epsilon_T(h, \hat{h}_S) - \epsilon_T(h, \hat{h}_T)|\right)\right),
\end{aligned}
\tag{25}
$$

where (a) is the outcome of applying the union bound Blitzer et al. (2007) with coefficient $\frac{N_T'}{N_S + N_T'}, \frac{N_S}{N_S + N_T'}$ respectively; (b) and (c) result from the Cauchy-Schwartz inequality Yin et al. (2024b) and (c) additionally adopt the assumption $N_T' \ll N_S$ following the sleight-of-hand Li et al. (2021).

Due to the samples are selected with high confidence, thus, we have the following assumption:

$$
\hat{\epsilon}_T \leq \epsilon_T \leq \hat{\epsilon}_S(h, \hat{h}) + \sqrt{\frac{4d}{N_S} \log(\frac{eN_S}{d}) + \frac{1}{N_S} \log(\frac{1}{\delta})} + 2C_f C_g W_1(\mathbb{P}_S(G), \mathbb{P}_T(G)) + \omega',
\tag{26}
$$

where $\omega' = \min_{||g||_{Lip} \le C_g, ||f||_{Lip} \le C_f} \{\epsilon_S(h, \hat{h}) + \epsilon_T(h, \hat{h})\}$, $\hat{\epsilon}_T$ is the empirical risk on the high confidence samples, $\epsilon_S$ is the empirical risk on the target domain. Besides, we have:

$$\min(|\epsilon_S(h, \hat{h}_S) - \epsilon_S(h, \hat{h}_T)|, |\epsilon_T(h, \hat{h}_S) - \epsilon_T(h, \hat{h}_T)|) \\ \le \min(\epsilon_S(h, \hat{h}_S) + \epsilon_T(h, \hat{h}_S)). \tag{27}$$

Therefore, we can derive the upper bound of the target domain generalization error $\epsilon_T(h, \hat{h}_T)$ as follows:

$$\epsilon_T(h, \hat{h}_T) \le \frac{N'_T}{N_S + N'_T} \hat{\epsilon}_T(h, \hat{h}_T) + \frac{N_S}{N_S + N'_T} \left( \hat{\epsilon}_S(h, \hat{h}_S) + \sqrt{\frac{4d}{N_S} \log(\frac{eN_S}{d}) + \frac{1}{N_S} \log(\frac{1}{\delta})} \right)$$
$$+ \frac{N_S}{N_S + N'_T} \left( 2C_f C_g W_1(\mathbb{P}_S(G), \mathbb{P}_T(G)) + \omega \right) \le \hat{\epsilon}_S(h, \hat{h}) \tag{28}$$
$$+ \sqrt{\frac{4d}{N_S} \log(\frac{eN_S}{d}) + \frac{1}{N_S} \log(\frac{1}{\delta})} + 2C_f C_g W_1(\mathbb{P}_S(G), \mathbb{P}_T(G)) + \omega'.$$

## 4.2 Proof of Theorem 2

**Proof of Theorem 2.** Assume the input space is $\mathcal{X} \subseteq \mathbb{R}^d$ and the label space is $\mathcal{Y} = \{1, 2, ..., K\}$. The real data distribution is $\mathcal{D}$, and the observation distribution with noisy labels is $\tilde{D}$, where the upper limit of the noise ratio is $\eta \le 0.5$. In the early stage of training (the first $T_0$ steps), the model's prediction of clean samples satisfies $p^i_{\theta_i}(x) \approx y_i$, where $t \le T_0$, and $(x, y)$ comes from the clean data subset $\mathcal{D}_{\text{clean}} \subset \hat{\mathcal{D}}_2$, we then have:

$$p_\theta(y_i|x_i) \approx \frac{\tilde{y}_i}{\tilde{y}_i + \lambda y_i(1 - p_\theta(y_i|x_i))}. \tag{29}$$

*Proof.* For clean samples $(x_i, y_i)$, the early EMA prediction satisfies $p^i \to y_i$ when the training step number $t \to \infty$. Decompose the loss into the contribution of clean samples and noise samples:

$$\mathcal{L}_{\text{CLS}} = \mathcal{L}_{\text{CE}}^{\text{clean}} + \mathcal{L}_{\text{CE}}^{\text{noisy}} + \lambda \left( \mathcal{L}_{\text{Reg}}^{\text{clean}} + \mathcal{L}_{\text{Reg}}^{\text{noisy}} \right). \tag{30}$$

For clean samples $(x_i, y_i)$, since $p^i \approx y_i$, the regularization term is approximately:

$$\mathcal{L}_{\text{Reg}}^{\text{clean}} \approx \frac{1}{N_{\text{clean}}} \sum_{i \in \text{clean}} \log \left( 1 - \langle p_\theta(y|x_i), y_i \rangle \right). \tag{31}$$

Due to $\langle p_\theta(y|x_i), y_i \rangle = p_\theta(y_i|x_i)$, when $p_\theta(y_i|x_i) \to 1$, $\log(1 - p_\theta(y_i|x_i)) \to -\infty$, but in actual optimization, through gradient descent, the model will adjust $p_\theta(y_i|x_i)$ to balance the cross entropy and regularization terms. For clean samples, the gradient of the regularization term is:

$$\nabla_\theta \mathcal{L}_{\text{Reg}}^{\text{clean}} \propto - \frac{\hat{p}_i}{1 - \langle p_\theta, \hat{p}_i \rangle} \cdot \nabla_\theta p_\theta. \tag{32}$$

Since $p^i \approx y_i$, the gradient direction encourages $p_\theta(y_i|x_i)$ to be close to $y_i$, consistent with the cross-entropy objective.

For the noise sample $(x_i, \tilde{y}_i \ne y_i)$, assuming that $p^i$ does not converge to any fixed distribution (it may be close to a uniform distribution or the wrong category):

$$\mathcal{L}_{\text{Reg}}^{\text{noisy}} = \frac{1}{N_{\text{noisy}}} \sum_{i \in \text{noisy}} \log \left( 1 - \langle p_\theta(y|x_i), \hat{p}_i \rangle \right). \tag{33}$$

If $\hat{p}_i$ is close to uniform distribution, then $\langle p_\theta, \hat{p}_i \rangle \approx \frac{1}{K}$, and the regularization term has little effect on the gradient; if $hatp_i$ is biased towards the wrong label, the regularization term prevents $p_\theta$ from overfitting $\hat{y}^i$.

The cross entropy gradient $\nabla_\theta \mathcal{L}_{\mathrm{CE}}^{\mathrm{noisy}}$ of the noise sample points in the wrong direction, while the regularization gradient $\nabla_\theta \mathcal{L}_{\mathrm{Reg}}^{\mathrm{noisy}}$ points in the opposite direction, thereby partially offsetting the effect of the noise. Under the assumption that $\hat{p}_i \to y_i$, the total loss of CLS is approximately:

$$\mathcal{L}_{\mathrm{CLS}} \approx \mathcal{L}_{\mathrm{CE}}^{\mathrm{clean}} + \lambda \mathcal{L}_{\mathrm{Reg}}^{\mathrm{clean}} + \eta. \tag{34}$$

Since the noise interference term is suppressed by the regularization term, the optimization process is mainly driven by clean samples. Assuming that the parameter $\theta$ is updated using gradient descent with a step size of $\beta$, the parameter update is:

$$\theta_{t+1} = \theta_t - \beta \nabla_\theta \mathcal{L}_{\mathrm{CLS}}. \tag{35}$$

When clean samples dominate, the gradient direction tends to minimize the true cross entropy $\mathcal{L}_{\mathrm{CE}}^{\mathrm{clean}}$. Since $p^i \approx y_i$, the gradient of the regularization term is approximately:

$$\nabla_\theta \mathcal{L}_{\mathrm{Reg}} \approx -\frac{1}{N_{\mathrm{clean}}} \sum_{i \in \mathrm{clean}} \frac{y_i}{1 - p_\theta(y_i | x_i)} \nabla_\theta p_\theta(y_i | x_i). \tag{36}$$

Combined with the cross entropy gradient $\nabla_\theta \mathcal{L}_{\mathrm{CE}} = -\frac{1}{N} \sum_{i=1}^{N} \frac{\tilde{y}_i}{p_\theta(y_i | x_i)} \nabla_\theta p_\theta(y_i | x_i)$:

$$p_\theta(y_i | x_i) \approx \frac{\tilde{y}_i}{\tilde{y}_i + \lambda y_i (1 - p_\theta(y_i | x_i))}, \tag{37}$$

When $\lambda$ is moderate, the model predicts $p_\theta(y_i \mid x_i)$ close to 1, consistent with the real label.

### 4.3   Proof of Theorem 3

**Theorem 3.** Assume that the model complexity is characterized by Rademacher complexity $\mathfrak{R}_n(\mathcal{F})$ Yin et al. (2019). For any $\delta > 0$, the generalization error upper bound of $\mathcal{L}_{\mathrm{CLS}}$ satisfies with probability $1 - \delta$:

$$\mathrm{GenError}_{\mathcal{L}_{\mathrm{CLS}}} \leq \mathrm{GenError}_{\mathcal{L}_{\mathrm{CE}}} \leq \frac{2\mathfrak{R}_n(\mathcal{F})}{\sqrt{\lambda}} + \sqrt{\frac{\log(1/\delta)}{2n}} + O\left(\frac{\eta + \epsilon}{\mu}\right). \tag{38}$$

*Proof.* For a function class $\mathcal{G}$, its Rademacher complexity is defined as:

$$\mathfrak{R}_N(\mathcal{G}) = \mathbb{E}_{x_i, \sigma_i} \left[ \sup_{g \in \mathcal{G}} \frac{1}{N} \sum_{i=1}^{N} \sigma_i g(x_i) \right]. \tag{39}$$

where $\sigma_i$ is an independent uniformly distributed Rademacher random variable. According to statistical learning theory, for the loss function $\ell$, the upper bound of the generalization error can be expressed as:

$$\mathrm{GenError} \leq 2\mathfrak{R}_N(\ell \circ \mathcal{F}) + \mathcal{O}\left(\sqrt{\frac{\log(1/\delta)}{N}}\right). \tag{40}$$

where $\ell \circ \mathcal{F} = \{(x, y) \mapsto \ell(f_\theta(x), y) \mid f_\theta \in \mathcal{F}\}$. In cross entropy loss $\mathcal{G}_{\mathrm{CE}} = \{(x, y) \mapsto \ell_{\mathrm{CE}}(f_\theta(x), y)\}$ and $\mathcal{L}_{\mathrm{CLS}}$ loss $\mathcal{G}_{\mathrm{ELR}} = \{(x, y) \mapsto \ell_{\mathrm{CE}}(f_\theta(x), y) + \lambda \mathcal{L}_{\mathrm{Reg}}(f_\theta)\}$, since $\mathcal{L}_{\mathrm{Reg}}$ introduces constraints on prediction consistency, the hypothesis space $\mathcal{F}_{\mathrm{CLS}}$ is more restricted than $\mathcal{F}_{\mathrm{CE}}$, that is:

$$\mathcal{F}_{\mathrm{CLS}} \subset \mathcal{F}_{\mathrm{CE}}. \tag{41}$$

Due to the inclusion relationship of the function class, its Rademacher complexity satisfies:

$$\mathfrak{R}_N(\mathcal{G}_{\mathrm{CLS}}) \leq \mathfrak{R}_N(\mathcal{G}_{\mathrm{CE}}). \tag{42}$$

Assuming $\ell_{\mathrm{CE}}$ is $L$-Lipschitz continuous and $\mathcal{L}_{\mathrm{Reg}}$ is $L'$-Lipschitz continuous, then we have:

$$\mathfrak{R}_N(\mathcal{G}_{\mathrm{CLS}}) \leq \mathfrak{R}_N(\mathcal{G}_{\mathrm{CE}}) + \lambda \cdot \mathfrak{R}_N(\mathcal{L}_{\mathrm{Reg}} \circ \mathcal{F}). \tag{43}$$

However, since the design goal of $\mathcal{L}_{\text{Reg}}$ is to constrain the consistency of model predictions (i.e., reduce variance), in practice $\mathfrak{R}_N(\mathcal{G}_{\text{CE}}$ grows slower than the complexity reduction of the cross entropy loss, resulting in lower overall complexity.

In the presence of noisy labels, the relationship between the true risk $R(f)$ and the empirical risk $\hat{R}_N(f)$ needs to be modified to:

$$R(f) \le \hat{R}_N(f) + 2\mathfrak{R}_N(\mathcal{G}) + 3\sqrt{\frac{\log(2/\delta)}{2N}} + \eta \cdot C, \tag{44}$$

where $C$ is a constant related to label noise. Combined with the complexity difference $\mathfrak{R}_N(\mathcal{G}_{\text{CLS}}) \le \mathfrak{R}_N(\mathcal{G}_{\text{CE}})$, and

$$\text{Error}_{\text{CLS}} \le 2\mathfrak{R}_N(\mathcal{G}_{\text{CLS}}) + \mathcal{O}\left(\sqrt{\frac{\log(1/\delta)}{N}}\right) + \eta \cdot C_{\text{CLS}},$$

$$\text{Error}_{\text{CE}} \le 2\mathfrak{R}_N(\mathcal{G}_{\text{CE}}) + \mathcal{O}\left(\sqrt{\frac{\log(1/\delta)}{N}}\right) + \eta \cdot C_{\text{CE}}. \tag{45}$$

Therefore, we can infer that the generalization error upper bound of $\mathcal{L}_{\text{CLS}}$ is lower.

# 5 Experiments

## 5.1 Experimental Setup

**Datasets.** IEMOCAP Busso et al. (2008) and MELD Poria et al. (2018) are commonly used multimodal databases in MERC. The IEMOCAP dataset includes 10 actors (5 men and 5 women). Each pair of actors simulates a real dialogue scene and conducts 5 conversations of about 1 hour, totaling about 12 hours. All conversations are manually annotated by emotion category. The MELD dataset is an extension of the EmotionLines dataset, and is designed for MERC. The dataset contains about 13,000 conversations, including more than 1,400 multi-turn conversations and about 13,000 single-turn conversations with emotion labels. All conversations are performed by actors and the scenes are set in the plot of the TV series. These datasets come from different scenarios and therefore represent a variety of different application areas. Following previous studies Zhang et al. (2024b), we selected samples of four common emotions: neutral, joy, sadness, and anger.

**Baselines.** To verify the superior performance of our proposed method DGDA, we compared it with other comparison methods, including traditional methods, i.e., TextCNN Kim (2014), LSTM Poria et al. (2017), DialogueRNN Majumder et al. (2019), MMGCN Hu et al. (2021), M3NET Chen et al. (2023), CFN-ESA Li et al. (2024a), SDT Ma et al. (2024), EmotionIC Liu et al. (2023), and DEDNet Wang et al. (2024a), denoising methods, i.e., OMG Yin et al. (2023), and SPORT Yin et al. (2024a), and domain adaption (DA) methods, i.e., A2GNN Liu et al. (2024b), Amanda Zhang et al. (2024b), and Boomda Sun et al. (2026).

**Implementation details.** All experiments are conducted in Python 3.9, using the PyTorch 2.1 framework, and computed on a NVIDIA A100 40GB GPU. We chose the Adam optimizer to train the model, and the initial learning rate is set to 0.0005. The loss function consisted of the cross entropy loss and the regularization loss we proposed, which prevented the model from memorizing noisy labels during training and thus improved the generalization ability of the model. The batch size is set to 32. In all experiments, the reported results are the average of 10 independent runs. The weight initialization of each run was random. To further evaluate the statistical significance of the experimental results, paired $t$-tests are performed on the results of the 10 runs. All $t$-test results show $p$ values less than 0.05, indicating that the differences in model performance in multiple experiments were statistically significant.

**Evaluation metrics.** For the multi-emotional dialogue datasets IEMOCAP and MELD, we adopt the **Weighted F1-score (WF1)** as the primary evaluation metric. Due to the inherent class imbalance in these datasets, WF1 provides a more reliable assessment than Macro-F1 by weighting each category according to its sample proportion.

Table 1: The performance of different methods is shown under different noisy rates on the IEMOCAP and MELD datasets. The arrow → means from source to target domains. Underlines indicate suboptimal performance. **Bold** results indicate the best performance. We set the noise rate to 10%.

| Methods | IEMOCAP → MELD | | | | | MELD → IEMOCAP | | | | |
|---|---|---|---|---|---|---|---|---|---|---|
| | Joy | Sadness | Neutral | Anger | WF1 | Joy | Sadness | Neutral | Anger | WF1 |
| ***Tradition*** | | | | | | | | | | |
| TextCNN | 38.79 | 14.47 | 54.07 | 29.98 | 44.12 | 39.13 | 60.65 | 35.52 | 54.21 | 46.73 |
| LSTM | 11.17 | 6.86 | 57.27 | 33.37 | 39.98 | 51.27 | 60.89 | 46.76 | 46.74 | 50.68 |
| DialogueRNN | 24.02 | 12.95 | 63.69 | 35.23 | 47.08 | 37.75 | 56.47 | 14.91 | 59.92 | 39.46 |
| MMGCN | 24.87 | 0.00 | 46.98 | 28.73 | 35.77 | 50.95 | 0.00 | 54.90 | 64.13 | 43.94 |
| M3NET | 45.15 | 4.69 | 38.16 | 29.25 | 35.48 | 46.91 | **71.08** | 35.31 | 71.76 | 54.72 |
| CFN-ESA | 9.90 | 3.23 | 69.36 | 3.16 | 42.08 | 19.26 | 9.38 | 44.62 | 16.50 | 25.57 |
| SDT | 8.74 | 6.78 | 67.10 | **40.06** | 45.96 | 50.18 | 61.20 | 50.25 | 64.86 | 56.58 |
| EmotionIC | 0.21 | **24.16** | 45.68 | 18.52 | 30.58 | 30.55 | 37.11 | 37.64 | 33.74 | 35.51 |
| DEDNet | 25.99 | 1.77 | 57.86 | 27.31 | 42.13 | 40.40 | 0.00 | 29.68 | 33.61 | 25.28 |
| ***Denoising*** | | | | | | | | | | |
| OMG | 34.73 | 4.56 | 58.19 | 31.23 | 44.92 | 47.55 | 67.21 | 41.23 | 54.22 | 51.54 |
| SPORT | 29.57 | 9.04 | 60.74 | 32.38 | 45.84 | 49.11 | 61.23 | 39.18 | 66.89 | 52.89 |
| ***Adaption*** | | | | | | | | | | |
| A2GNN | 41.24 | 3.91 | 66.73 | 27.15 | 50.50 | 53.11 | 55.43 | 50.18 | 60.38 | 54.46 |
| Amanda | 36.29 | 10.15 | 62.18 | 29.03 | 47.69 | 55.68 | 45.81 | 54.28 | 56.76 | 53.14 |
| Boomda | 45.69 | 6.87 | 65.19 | 31.42 | 51.40 | 60.19 | 44.19 | 51.01 | 69.77 | 55.57 |
| DGDA | **56.21** | 9.81 | **76.68** | 36.00 | **60.99** | **61.04** | 66.10 | **57.19** | **82.91** | **66.47** |

Table 2: The performance of different methods is shown under different noisy rates on the IEMOCAP and MELD datasets. The arrow → means from source to target domains. **Bold** results indicate the best performance. Underlines indicate suboptimal performance. We set the noise rate to 20%.

| Methods | IEMOCAP → MELD | | | | | MELD → IEMOCAP | | | | |
|---|---|---|---|---|---|---|---|---|---|---|
| | Joy | Sadness | Neutral | Anger | WF1 | Joy | Sadness | Neutral | Anger | WF1 |
| ***Tradition*** | | | | | | | | | | |
| TextCNN | 38.56 | 13.11 | 51.71 | 29.19 | 42.51 | 36.46 | 54.51 | 32.58 | 52.26 | 43.33 |
| LSTM | 10.76 | 5.70 | 54.03 | 32.34 | 37.82 | 45.85 | 55.67 | 43.47 | 43.85 | 46.74 |
| DialogueRNN | 22.82 | 11.40 | 60.06 | 34.34 | 44.52 | 35.04 | 51.49 | 14.60 | 55.60 | 36.68 |
| MMGCN | 22.62 | 0.00 | 45.54 | 27.99 | 34.38 | 46.66 | 0.00 | 53.57 | 59.24 | 41.59 |
| M3NET | 41.50 | 5.45 | 33.53 | 26.18 | 31.74 | 42.24 | **64.74** | 32.60 | 65.40 | 49.94 |
| CFN-ESA | 9.13 | 7.14 | **69.07** | 3.40 | 42.13 | 14.39 | 19.65 | 41.20 | 29.47 | 29.38 |
| SDT | 8.75 | 5.22 | 62.88 | 37.65 | 43.10 | 44.10 | 53.77 | 48.23 | 60.01 | 52.00 |
| EmotionIC | 1.80 | 21.76 | 39.57 | 14.98 | 26.75 | 33.12 | 34.81 | 17.77 | 44.26 | 30.76 |
| DEDNet | 25.21 | 1.40 | 54.80 | 26.52 | 40.09 | 42.24 | 0.00 | 39.37 | 12.19 | 30.76 |
| ***Denoising*** | | | | | | | | | | |
| OMG | 27.65 | 5.51 | 53.23 | 26.22 | 40.02 | 42.74 | 61.89 | 35.66 | 53.51 | 47.39 |
| SPORT | 23.58 | 13.55 | 55.73 | 27.87 | 41.52 | 44.17 | 56.38 | 33.47 | 70.69 | 49.95 |
| ***Adaption*** | | | | | | | | | | |
| A2GNN | 36.23 | 7.11 | 61.24 | 32.65 | 47.38 | 46.28 | 48.59 | 51.93 | 54.88 | 51.13 |
| Amanda | 30.83 | 5.14 | 57.67 | 24.83 | 42.97 | 51.33 | 40.53 | 47.31 | 52.27 | 47.58 |
| Boomda | 49.27 | 12.38 | 60.77 | 26.42 | 49.37 | 54.63 | 38.91 | 45.73 | 74.49 | 52.84 |
| DGDA | **54.97** | **25.61** | 66.11 | **48.76** | **57.87** | **59.60** | 40.74 | **64.28** | **77.48** | **61.56** |

Table 3: The performance of different methods is shown under different noisy rates on the IEMOCAP and MELD datasets. The arrow → means from source to target domains. **Bold** results indicate the best performance. Underlines indicate suboptimal performance. We set the noise rate to 30%.

| Methods | IEMOCAP → MELD | | | | | MELD → IEMOCAP | | | | |
| --- | --- | --- | --- | --- | --- | --- | --- | --- | --- | --- |
| | Joy | Sadness | Neutral | Anger | WF1 | Joy | Sadness | Neutral | Anger | WF1 |
| *Tradition* | | | | | | | | | | |
| TextCNN | 34.59 | 9.87 | 47.79 | 26.69 | 38.83 | 35.46 | 52.43 | 30.87 | 47.59 | 40.87 |
| LSTM | 9.74 | 4.57 | 49.49 | 29.03 | 34.48 | 43.19 | 51.54 | 40.71 | 38.57 | 43.03 |
| DialogueRNN | 20.81 | 8.62 | 54.72 | **31.3** | 40.41 | 34.10 | 49.35 | 13.84 | 49.94 | 34.31 |
| MMGCN | 21.64 | 0.00 | 41.58 | 25.73 | 31.62 | 42.27 | 0.00 | **50.85** | 53.90 | 38.59 |
| M3NET | 40.43 | 7.58 | 31.80 | 23.16 | 30.30 | 39.68 | 60.41 | 31.84 | 59.79 | 46.84 |
| CFN-ESA | 18.25 | 3.38 | **67.01** | 0.61 | 42.15 | 14.53 | 20.45 | 45.78 | 15.42 | 27.63 |
| SDT | 7.61 | 4.24 | 56.37 | 31.16 | 38.20 | 40.84 | 48.39 | 44.76 | 55.03 | 47.72 |
| EmotionIC | 4.85 | 17.87 | 19.14 | 8.34 | 14.56 | 26.60 | 39.60 | 22.16 | 31.23 | 29.20 |
| DEDNet | 23.87 | 1.09 | 49.70 | 22.6 | 36.35 | 39.05 | 0.00 | 37.85 | 13.57 | 22.90 |
| *Denoising* | | | | | | | | | | |
| OMG | 22.44 | 3.14 | 48.47 | 22.87 | 35.57 | 40.15 | 57.61 | 32.78 | 50.06 | 44.08 |
| SPORT | 18.37 | 8.59 | 53.16 | 22.83 | 37.85 | 42.39 | 52.18 | 30.17 | 51.19 | 42.46 |
| *Adaption* | | | | | | | | | | |
| A2GNN | 34.47 | 2.82 | 54.31 | 29.48 | 42.27 | 42.17 | 43.12 | 45.69 | 49.37 | 45.55 |
| Amanda | 26.88 | 3.83 | 57.96 | 20.01 | 41.54 | 47.33 | 34.49 | 46.14 | 50.15 | 44.63 |
| Boomda | 46.13 | 7.18 | 54.48 | 21.36 | 44.06 | **51.13** | 39.17 | 42.23 | **70.85** | 50.18 |
| DGDA | **57.16** | **24.22** | 65.99 | 21.36 | **54.36** | 49.82 | **62.96** | 47.06 | 68.79 | **56.79** |

Table 4: The performance of different methods is shown under different noisy rates on the IEMOCAP and MELD datasets. The arrow → means from source to target domains. **Bold** results indicate the best performance. Underlines indicate suboptimal performance. We set the noise rate to 40%.

| Methods | IEMOCAP → MELD | | | | | MELD → IEMOCAP | | | | |
| --- | --- | --- | --- | --- | --- | --- | --- | --- | --- | --- |
| | Joy | Sadness | Neutral | Anger | WF1 | Joy | Sadness | Neutral | Anger | WF1 |
| *Tradition* | | | | | | | | | | |
| TextCNN | 33.92 | 9.04 | 44.59 | 24.53 | 36.51 | 34.33 | 48.97 | 26.65 | 43.49 | 37.29 |
| LSTM | 9.67 | 4.61 | 44.72 | 26.50 | 31.41 | **41.46** | 48.47 | 36.83 | 37.65 | 40.40 |
| DialogueRNN | 20.60 | 8.08 | 49.99 | 28.84 | 37.30 | 33.55 | 47.75 | 12.01 | 44.78 | 31.85 |
| MMGCN | 21.46 | 0.00 | 38.70 | 23.59 | 29.66 | 40.82 | 0.00 | **46.71** | 51.12 | 36.14 |
| M3NET | 39.99 | 6.48 | 28.77 | 20.79 | 28.07 | 38.09 | 55.89 | 27.35 | 54.40 | 42.52 |
| CFN-ESA | 10.47 | 5.00 | **68.76** | 4.20 | 42.16 | 10.19 | 20.05 | 46.56 | 25.02 | 29.71 |
| SDT | 7.34 | **36.10** | 50.78 | **29.34** | 37.49 | 38.01 | 44.88 | 40.09 | 49.54 | 43.36 |
| EmotionIC | 37.53 | 11.72 | 2.11 | 9.43 | 11.34 | 24.74 | 33.33 | 15.07 | 30.98 | 24.80 |
| DEDNet | 23.14 | 0.92 | 44.84 | 20.68 | 33.17 | 37.64 | 0.00 | 36.42 | 11.59 | 21.67 |
| *Denoising* | | | | | | | | | | |
| OMG | 20.58 | 6.14 | 45.23 | 19.97 | 33.21 | 38.83 | 53.31 | 29.91 | 47.42 | 41.15 |
| SPORT | 15.52 | 4.17 | 46.31 | 18.47 | 32.39 | 38.74 | 50.15 | 27.14 | 46.68 | 39.19 |
| *Adaption* | | | | | | | | | | |
| A2GNN | 31.25 | 4.63 | 52.29 | 26.14 | 40.15 | 41.43 | 40.05 | 42.22 | 46.38 | 42.68 |
| Amanda | 41.28 | 9.16 | 50.06 | 16.77 | 40.08 | 41.39 | 36.62 | 36.12 | 42.34 | 38.59 |
| Boomda | **41.29** | 3.33 | 50.02 | 15.53 | 39.38 | 36.07 | 34.28 | 41.27 | 44.51 | 39.75 |
| DGDA | 37.85 | 13.51 | 64.74 | 28.91 | **49.74** | 35.58 | **56.90** | 35.51 | **57.51** | **46.21** |

Table 5: The ablation studies are shown under different noisy rates on the IEMOCAP and MELD datasets. The arrow → means from source to target domains. **Bold** results indicate the best performance. We set the noise rate to 10%.

| Methods | IEMOCAP → MELD | | | | | MELD → IEMOCAP | | | | |
|---|---|---|---|---|---|---|---|---|---|---|
| | Joy | Sadness | Neutral | Anger | WF1 | Joy | Sadness | Neutral | Anger | WF1 |
| DGDA-HGNN | 40.17 | 11.41 | 69.41 | 32.58 | 53.19 | 56.46 | 54.99 | 42.79 | 77.15 | 56.45 |
| DGDA-PathNN | 43.59 | 9.87 | 62.43 | 31.68 | 49.69 | 52.47 | 48.31 | 50.07 | 67.43 | 54.49 |
| DGDA/$\sigma^{HGNN}$ | 44.39 | 7.78 | 74.95 | 20.48 | 55.23 | 58.44 | 57.03 | 45.67 | 73.41 | 57.29 |
| DGDA/$\sigma^{PathNN}$ | 48.37 | 7.65 | 75.27 | 18.93 | 56.01 | 56.55 | 62.39 | 51.65 | 74.83 | 60.85 |
| DGDA/AP | 28.96 | 4.25 | 66.79 | 25.27 | 47.74 | 34.97 | 42.53 | 54.78 | 58.09 | 50.01 |
| DGDA/BC | 19.69 | **13.74** | 60.03 | 30.02 | 43.46 | 30.05 | 54.17 | 46.28 | 47.89 | 46.26 |
| DGDA/RT | 51.33 | 6.77 | 71.29 | **37.18** | 56.82 | 56.88 | 65.76 | 52.19 | 77.97 | 62.69 |
| DGDA | **56.21** | 9.81 | **76.68** | 36.00 | **60.99** | **61.04** | **66.10** | **57.19** | **82.91** | **66.47** |

First, for each emotion class $i$, the F1-score is computed as the harmonic mean of precision and recall:

$$F_i = \frac{2 \cdot (\text{Precision}_i \cdot \text{Recall}_i)}{\text{Precision}_i + \text{Recall}_i}, \tag{46}$$

where $\text{Precision}_i$ measures the correctness of predictions for class $i$, and $\text{Recall}_i$ measures how many true samples of class $i$ are correctly identified.

The overall WF1 is then obtained by weighting each class-specific $F_i$ using its sample count $n_i$:

$$\text{WF1} = \sum_{i=1}^{N} \left( \frac{n_i}{\sum_{j=1}^{N} n_j} \cdot F_i \right), \tag{47}$$

where $N$ denotes the total number of emotion categories.

Compared with unweighted metrics, WF1 more faithfully reflects model performance under imbalanced distributions, prevents both minority and majority classes from dominating the overall score, and offers a stable and comprehensive evaluation by jointly considering precision and recall.

## 5.2 Comparison with the State-of-the-arts

Tables 1, 2, 3, and 4 show the performance comparison results of our proposed DGDA method and various baseline methods on the IEMOCAP and MELD datasets under different noise conditions. The following important findings can be observed through comparative analysis. First, DA methods, including A2GNN, Amanda, Boomda, and DGDA, are generally better than traditional methods and show more robust emotion recognition performance regardless of the noise interference conditions. This shows that the traditional methods are difficult to effectively model the distribution difference between the source domain and the target domain. Second, compared with various recently proposed DA methods (A2GNN, Amanda, Boomda) and typical denoising methods (OMG, SPORT), DGDA has achieved better performance. The performance improvement may be attributed to the synergy of the following key design factors: (i) First, DGDA adopts a dual-branch graph semantic extraction mechanism to model and integrate graph structure information based on message passing and shortest path aggregation strategies, respectively. This design effectively leverages the complementary advantages of the two models in local relationship modeling and global structure capture. (ii) Secondly, DGDA introduces a branch coupling module and an adaptive perturbation mechanism to dynamically adjust the interaction between the two branches, which not only promotes the efficient transfer of cross-domain knowledge but also effectively alleviates the negative impact of category distribution differences. (iii) Finally, we fully consider the two key factors of domain invariance and noise label interference. Through a joint optimization strategy, while ensuring feature domain alignment, we effectively suppress the negative impact of noisy labels on the model learning process.

Table 6: The ablation studies are shown under different noisy rates on the IEMOCAP and MELD datasets. The arrow → means from source to target domains. **Bold** results indicate the best performance. We set the noise rate to 20%.

| Methods | IEMOCAP → MELD | | | | | MELD → IEMOCAP | | | | |
|---|---|---|---|---|---|---|---|---|---|---|
| | Joy | Sadness | Neutral | Anger | WF1 | Joy | Sadness | Neutral | Anger | WF1 |
| DGDA-HGNN | 36.07 | 7.77 | 66.00 | 29.34 | 49.64 | 53.21 | 51.47 | 40.09 | 73.97 | 53.36 |
| DGDA-PathNN | 39.77 | 6.26 | 57.87 | 28.38 | 45.54 | 49.27 | 45.05 | 47.46 | 63.93 | 51.42 |
| DGDA/$\sigma^{HGNN}$ | 39.71 | 4.72 | 70.37 | 15.73 | 50.73 | 55.98 | 53.49 | 41.75 | 70.72 | 53.99 |
| DGDA/$\sigma^{PathNN}$ | 44.56 | 4.30 | **71.80** | 15.75 | 52.52 | 53.54 | 59.50 | 47.97 | 72.54 | 57.80 |
| DGDA/AP | 25.02 | 7.33 | 63.48 | 22.15 | 44.88 | 32.35 | 38.62 | 52.36 | 54.77 | 46.98 |
| DGDA/BC | 18.73 | 8.96 | 55.04 | 25.68 | 39.41 | 27.63 | 50.51 | 43.11 | 44.89 | 43.13 |
| DGDA/RT | 46.83 | 3.24 | 67.01 | 32.97 | 52.57 | 54.54 | **62.48** | 50.13 | 73.99 | 59.81 |
| DGDA | **54.97** | **25.61** | 66.11 | **48.76** | **57.87** | **59.60** | 40.74 | **64.28** | **77.48** | **61.56** |

Table 7: The ablation studies are shown under different noisy rates on the IEMOCAP and MELD datasets. The arrow → means from source to target domains. **Bold** results indicate the best performance. We set the noise rate to 30%.

| Methods | IEMOCAP → MELD | | | | | MELD → IEMOCAP | | | | |
|---|---|---|---|---|---|---|---|---|---|---|
| | Joy | Sadness | Neutral | Anger | WF1 | Joy | Sadness | Neutral | Anger | WF1 |
| DGDA-HGNN | 33.28 | 3.21 | 62.38 | 25.83 | 46.13 | 50.47 | 48.93 | 38.01 | 69.02 | 50.34 |
| DGDA-PathNN | 35.99 | 2.61 | 53.52 | 23.48 | 41.30 | 47.06 | 42.46 | 42.80 | 61.26 | 48.10 |
| DGDA/$\sigma^{HGNN}$ | 35.73 | 2.74 | 67.35 | 10.94 | 47.36 | 53.77 | 51.10 | 39.30 | 66.06 | 51.01 |
| DGDA/$\sigma^{PathNN}$ | 39.61 | 1.70 | **69.56** | 13.14 | 49.63 | **50.95** | 56.79 | 43.36 | **69.06** | 54.21 |
| DGDA/AP | 20.93 | 5.06 | 60.07 | 19.45 | 41.52 | 28.86 | 35.47 | **49.80** | 52.39 | 44.20 |
| DGDA/BC | 16.50 | 5.39 | 52.21 | 22.03 | 36.53 | 25.51 | 48.00 | 40.81 | 41.26 | 40.46 |
| DGDA/RT | 42.96 | 5.89 | 64.37 | **30.05** | 50.09 | 50.93 | 60.22 | 46.30 | 60.06 | 53.76 |
| DGDA | **57.16** | **24.22** | 65.99 | 21.36 | **54.36** | 49.82 | **62.96** | 47.06 | 68.79 | **56.79** |

## 5.3 Ablation Study

To comprehensively analyze the actual contribution of each module of DGDA to the overall model performance, we designed and conducted multiple sets of ablation experiments as shown in Tables 5, 6, 7, and 8. Specifically, we constructed the following seven variant configurations: (1) DGDA-HGNN: HGNN is used in both branches for graph semantic feature extraction; (2) DGDA-PathNN: PathNN is used in both branches. (3) DGDA/$\sigma^{HGNN}$: The perturbation module is removed from the HGNN branch of DGDA. (4) DGDA/$\sigma^{PathNN}$: The perturbation module is removed from the PathNN branch. (5) DGDA/AP: The perturbation module is removed from both branches. (6) DGDA/BC: The branch coupling module is removed. (7) DGDA/RT: The regularization term is removed from both branches. DGDA significantly outperforms DGDA-HGNN and DGDA-PathNN in overall performance, indicating that a single graph semantic modeling method is not sufficient to extract and fuse graph semantic features from different perspectives. DGDA shows significant performance advantages in comparison with the three versions of the perturbation removal module (DGDA/$\sigma^{HGNN}$, DGDA/$\sigma^{PathNN}$, and DGDA/AP). By introducing perturbations in the feature space, the model can effectively prevent overfitting of the source domain features. Due to the lack of coupling and category alignment mechanism between branches, DGDA/BC has a significantly insufficient ability to distinguish categories in the target domain. Furthermore, without the introduction of regularization loss, the overall performance of the model shows a certain degree of degradation when facing training data with noisy labels.

Table 8: The performance of different methods is shown under different noisy rates on the IEMOCAP and MELD datasets. The arrow → means from source to target domains. **Bold** results indicate the best performance. We set the noise rate to 40%.

| Methods | IEMOCAP → MELD | | | | | MELD → IEMOCAP | | | | |
|---|---|---|---|---|---|---|---|---|---|---|
| | Joy | Sadness | Neutral | Anger | WF1 | Joy | Sadness | Neutral | Anger | WF1 |
| DGDA-HGNN | 29.05 | 2.17 | 60.09 | 22.28 | 43.37 | 37.82 | 44.14 | 35.26 | 44.37 | 40.05 |
| DGDA-PathNN | 33.60 | 4.32 | 51.39 | 19.75 | 39.23 | 33.53 | 36.77 | 38.02 | 48.31 | 39.77 |
| DGDA/$\sigma^{HGNN}$ | 33.23 | 6.86 | 64.70 | 6.45 | 45.07 | 31.58 | 39.01 | 34.45 | **61.35** | 42.08 |
| DGDA/$\sigma^{PathNN}$ | 35.08 | 3.19 | 60.37 | 8.89 | 43.02 | **38.82** | 31.96 | 40.72 | 46.25 | 39.85 |
| DGDA/AP | 17.75 | 1.72 | 56.67 | 16.23 | 38.20 | 26.81 | 31.16 | **47.53** | 48.98 | 41.19 |
| DGDA/BC | 11.91 | 2.98 | 47.58 | 17.50 | 32.12 | 23.35 | 43.63 | 36.36 | 38.76 | 36.85 |
| DGDA/RT | **38.22** | 3.21 | 59.93 | 25.67 | 45.75 | 26.23 | 37.51 | 32.33 | 47.81 | 36.70 |
| DGDA | 37.85 | **13.51** | **64.74** | **28.91** | **49.74** | 35.58 | **56.90** | 35.51 | 57.51 | **46.21** |

## 5.4 Effect of Different Modalities

To further explore the contribution of different modalities in emotion recognition, we designed a modality ablation experiment under the experimental condition of 10% noise rate. By gradually removing one or two modalities, we observed the performance differences of the model under various combinations. Table 3 shows the results of different modal combinations. First, for the single-modal experimental results, the performance of the text modality is far better than that of the audio modality and the visual modality. This phenomenon shows that although multimodal information has a synergistic effect, text features are still the most critical basis for emotion discrimination. Secondly, in the dual-modal combination experiment, the performance of all combinations is better than the corresponding single-modal results, which verifies that there is a complementary relationship between different modalities and joint modeling helps to improve the emotion recognition effect. Finally, when the three modalities are simultaneously involved in feature modeling, the model achieves the best recognition effect, significantly better than all single-modal and bimodal configurations.

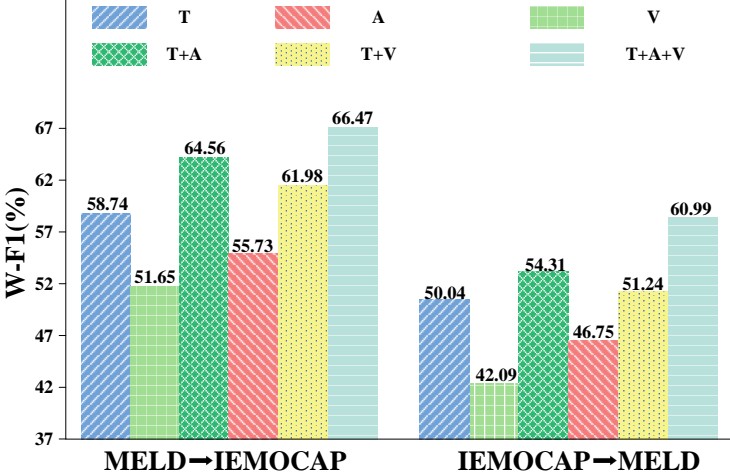

Figure 3: Verify the effectiveness of multimodal features.

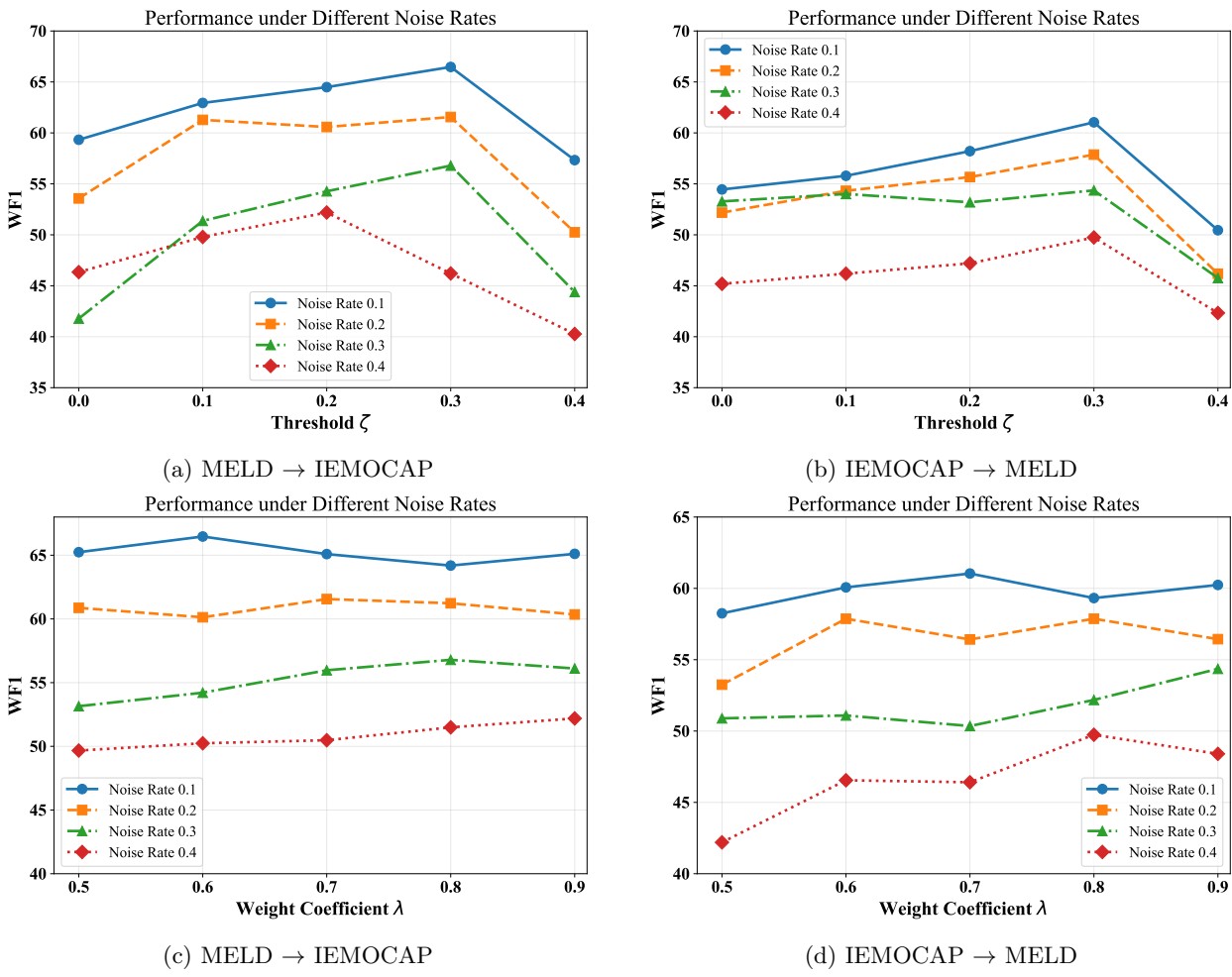

Figure 4: Hyperparameter sensitivity of threshold $\zeta$ and regularization weight $\lambda$.

## 5.5 Sensitivity Analysis

Figs. 4 (a) and (b) show the effect of the pseudo-label selection threshold $\zeta$ on model performance under different noise rates. When the noise rate is 0.1, as the threshold increases, the performance of the model increases significantly, and reaches the optimal value at 0.3, and then decreases slightly when the threshold is too high. This shows that under low-noise conditions, appropriately improving the pseudo-label selection criteria can help screen out more reliable pseudo-labels. However, when the threshold is further increased, the number of optional pseudo-label samples decreases significantly, resulting in insufficient supervision signals. Under high noise rates, the impact of threshold changes on model performance is relatively gentle. This is mainly because in a high-noise environment, the quality of the candidate pseudo-labels themselves is poor, and it is difficult to completely solve the pseudo-label noise problem by simply increasing the threshold. Figs. 4 (c) and (d) further analyze the impact of the regularization weight $\lambda$ on model performance. The overall trend shows that at lower $\lambda$, the model performance is poor, especially at high noise rates, and the WFI performance is low. As $\lambda$ gradually increases, the effect of the model at each noise rate is generally improved, and the best state is reached when $\lambda$ is about 0.7∼0.8. This shows that appropriately increasing the weight of the regularization term can better suppress the overfitting of the model to the noisy pseudo-labels, especially in a high-noise environment. However, it is worth noting that when $\lambda$ continues to increase to 0.9, the model performance under some noise rates decreases slightly.

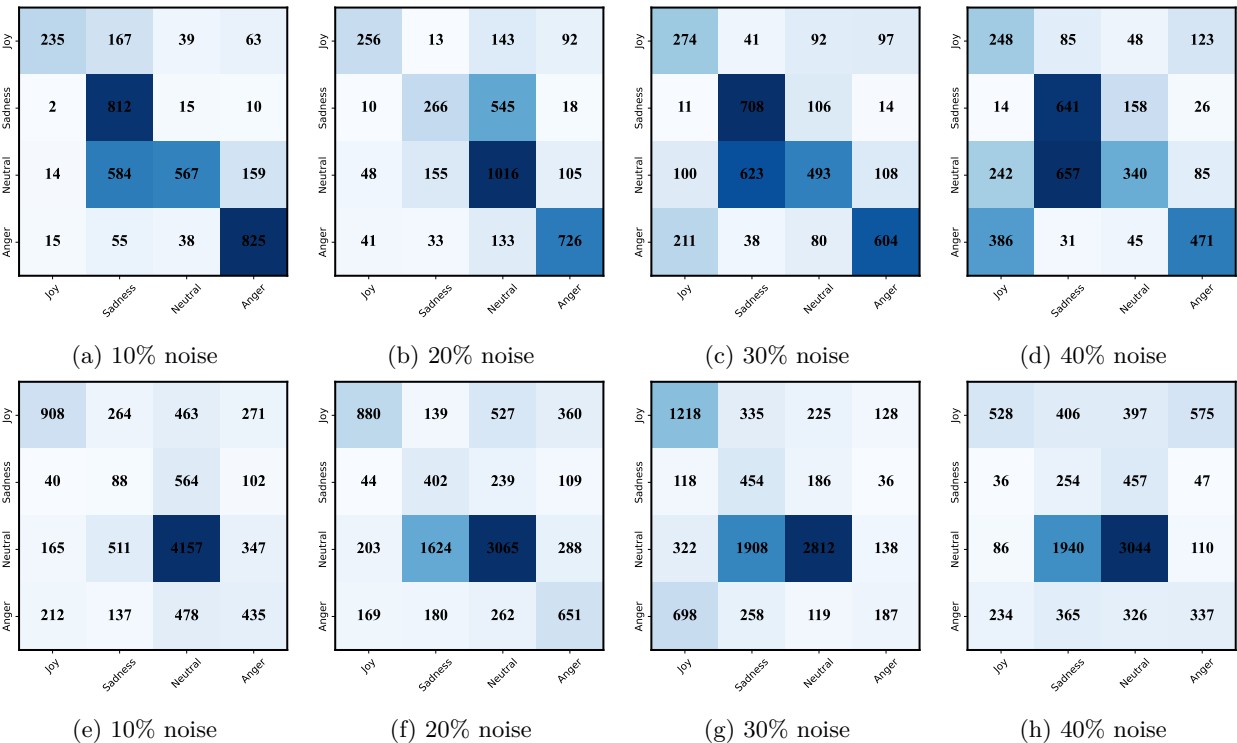

Figure 5: Confusion matrices for multimodal emotion recognition datasets. The matrices provide insights into the model's classification accuracy, highlighting the challenges and successes in distinguishing between different emotional categories. **Top:** Results with varying noise levels on the first dataset setting. **Bottom:** Results with varying noise levels on the second dataset setting.

## 5.6 Confusion Matrices

Fig. 5 shows the confusion matrix of the model prediction results on the IEMOCAP and MELD datasets, which provides an important basis for an in-depth understanding of the model's classification ability and performance differences in different emotion categories. It can be observed that the model has a relatively ideal recognition effect on the two categories of "Joy" and "Neutral". Most of the samples belonging to these two categories are accurately classified, and the misclassification ratio between the two is low. This phenomenon shows that the model has successfully learned the discriminative features that are highly related to these two categories of emotions and can effectively distinguish them, reflecting its good modeling ability for common emotion categories. However, the confusion matrix also reveals the classification difficulties in low-frequency emotion categories such as "Sadness" and "Anger". Compared with common emotions such as "Joy" and "Neutral", these categories already have the problem of insufficient sample number in multimodal emotion recognition datasets, and the category imbalance phenomenon is more significant. Due to the small number of "Sadness" and "Anger" samples available for learning during the training process, the model has certain difficulties in capturing the key patterns and features related to these two categories of emotions, resulting in its insufficient generalization ability on these two categories of samples. Therefore, the model's recognition effect on these two types of emotions is significantly inferior, and a high proportion of misclassification occurs. Specifically, the confusion matrix shows that a considerable number of samples in the "Sadness" and "Anger" categories are mistakenly classified into other categories such as "Happy" or "Neutral". This result reflects that the model has a certain degree of discrimination ambiguity between these emotions; that is, when distinguishing low-frequency emotion categories from common emotion categories, it may rely on overlapping or fuzzy features in emotional expressions, which leads to confusion.

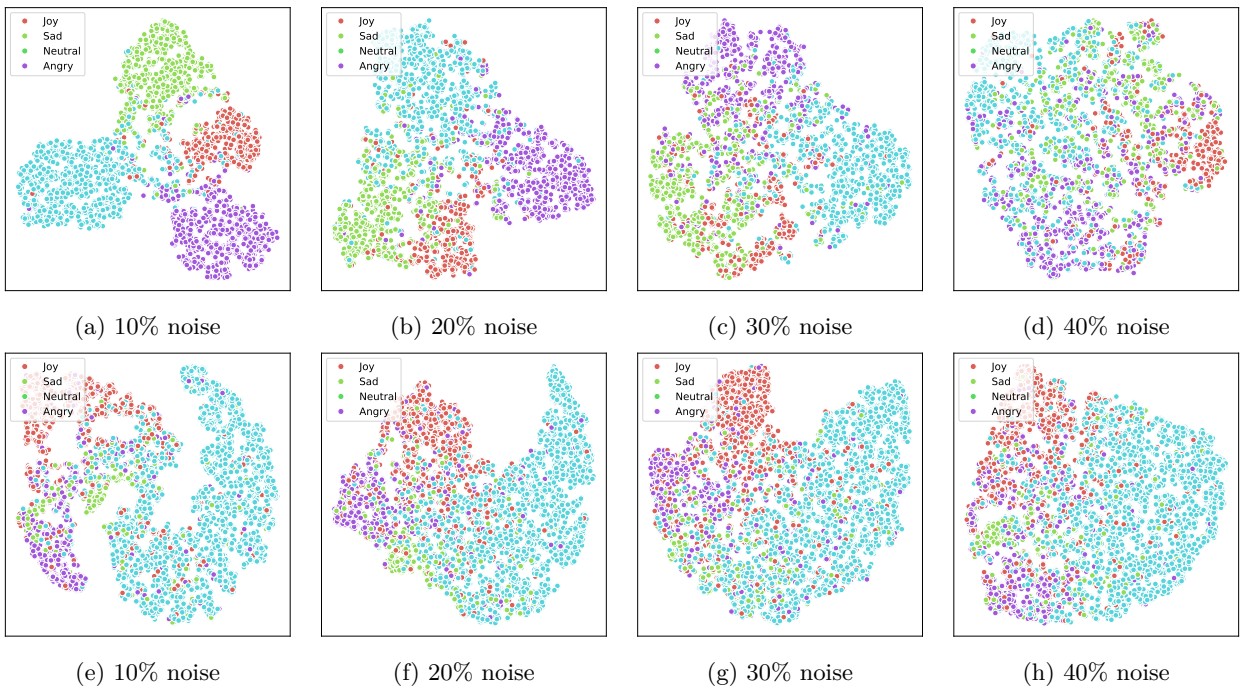

Figure 6: Visualization of the learned embeddings. **Top:** Trained on the MELD dataset and tested on the IEMOCAP dataset (with varying noise levels). **Bottom:** Trained on the IEMOCAP dataset and tested on the MELD dataset (with varying noise levels).

## 5.7 Visualization

Fig. 6 shows the distribution of emotional features learned by the model on the IEMOCAP and MELD datasets under different noise ratios (10%, 20%, 30%, and 40%), which intuitively reveals the impact of noise level changes on the model's discriminative ability and the effectiveness of the proposed method in combating noise interference. From the visualization results, on the IEMOCAP dataset (upper row in the figure), as the noise ratio gradually increases, the boundaries between different emotional categories begin to become blurred, the sample distribution within the category tends to be loose, and the feature overlap between categories becomes more obvious. Especially when the noise ratio reaches 30% and 40%, the degree of confusion between emotional categories increases significantly, and samples of categories such as Sad and Neutral, Joy and Angry appear to be distributed in a large area, resulting in a sharp decline in the model's ability to distinguish these emotional categories, and the reliability of the discrimination results is greatly affected. This phenomenon reflects that in a medium-to-high noise environment, the potential feature commonality between emotional categories and the perturbation effect of noise increase the learning difficulty of the model, making it difficult to maintain the effective extraction of discriminative features. In contrast, the MELD dataset (lower line in the figure) shows stronger robustness and category separability under the same noise level. Although the noise ratio also increased from 10% to 40%, the feature distribution between emotion categories still maintained a good clustering structure and relatively clear category boundaries. Even under the most stringent 40% noise condition, the samples of each category still showed a relatively stable distribution pattern, and the distinction between categories was preserved to a certain extent. This result fully demonstrates that the proposed method exhibits superior noise resistance on the MELD dataset, and can effectively suppress the erosion of the model feature space by erroneous labels in an environment with significant noise interference, ensuring that the model maintains good discrimination ability for emotion categories. It is worth noting that the MELD dataset itself is more challenging in terms of modal combination, context diversity, and speaker complexity. Therefore, the model can still show strong noise resistance on this

dataset, which further proves the robustness and generalization ability of the proposed method in the face of complex multimodal data and label uncertainty.

## 6 Conclusions

In this paper, we propose a Dual-branch Graph Domain Adaptation (DGDA) for multi-modal emotion recognition in cross-scenario conversations. Specifically, we first construct an emotion interaction graph to model the complex emotional dependencies between utterances. Then, we design a neighborhood aggregation and path aggregation dual-branch graph encoder to explicitly and implicitly capture the dynamic changes in emotion between utterances and explore multivariate relationships, respectively. To address the problem of out-of-domain distribution differences, we introduce a domain adversarial classifier to improve the representation ability of invariant features in the source domain. Furthermore, we construct a regularization loss to prevent the model from memorizing noise and improve the model's ability to resist interference from noisy labels.

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
