# OpenReview forum: "Dual-branch Graph Domain Adaptation for Cross-scenario Multi-modal Emotion Recognition"
_TMLR — Rejected by TMLR_

### Review · Reviewer_Mjyn · 2026-05-01

**Summary Of Contributions:**

Authors propose a new model DGDA for Unsupervised Domain Adaptation which combines the strengths of prior methods like HGNN, PathNN, co-training type loss, domain discriminative training, early-training regularization. Results are shown on MERC task with 2 datasets. Comparisons of results are provided with prior methods. Some ablation and analysis is also provided.

**Audience:**

Yes

**Audience Explanation:**

The choice of topic is apt for TMLR audience I think.

**Broader Impact Concerns:**

NIL

**Claims And Evidence:**

No

**Claims Explanation:**

I am not particularly convinced with how results are reported. WF1 metric can suppress minority-class contribution. This makes it hard to draw conclusions. macroF1 or UAR metrics should also have been reported I think along with standard deviations. Also, I dont see results for noise rate=0% baseline. I also have questions about the effectiveness of proposal since ablation experiments reveal that all components of the proposal are highly critical and removing even one component is taking performance of model to below baseline. This indicates the model is ad hoc combo of various tricks and not principled method with obvious merits. IMO model should not be this brittle. Nevertheless, the model still can be ok with major changes. The paper could have been organized better too. The proofs should be in appendix and various tables could be reduced in font size or compressed somehow. There are too many tables and it is hard to read and compare all of them. In the end, I think the training of the proposed model is novel contribution but that is also the biggest weakness as there is no pseudo-code of training algorithm. With so many loss functions and tricks spread out on various pages, it is unclear how model is trained. Clear pseudo-code is mandatory for this model IMO.

**Requested Changes:**

Major changes requested:
1. Pseudo-code of model training
2. Standard deviation in result tables
3. Clarification in PathNN: all nodes get updated as equation 4 suggests? or just start and end node as page 5 text suggests
4. Baseline results for noise rate=0%
5. Topline for supervised results on both datasets. I want to see the graph models are actually powerful as compared to speech foundation model fine-tuning. So, I want supervision models using GNNs, MLP, and FMs.
6. Discuss somewhere if pretrained GNNs makes sense to use in future work. I didnt find future work discussion
7. Why only target domain loss in Eq 7
8. Why perturbation is so important for your model
9. How IEMOCAP is suitable for contextual emotion recognition when its labels are not contextual? (i think the labels were annotated per utterance independently) I may be wrong or not understanding MERC task definition. Anyways, this needs clarification for readers.
10. unimodal comparisons with previous works (which also use unimodal for fairness).
11. In Sec.5 "Baselines", previous works are simply listed. Perhaps in related work section, include brief description of those methods.
12. Section 5.1 last para is not factual. WF1 does have limitations. It is not absolute best metric
13. Cross-domain and cross-scenario terms are interchangeably used. Not sure it is a right thing
14. Move proofs to appendix and compress tables somehow. Maybe a graph which compares all noise rates just like you have done in Fig 4

---

### Review · Reviewer_f2mH · 2026-05-07

**Summary Of Contributions:**

The paper proposes an architecture for multimodal emotion recognition. It is designed to address domain shift and noisy label interference by incorporating a dual-branch encoder and a domain-adversarial alignment strategy. The method is evaluated through multiple experiments, mainly focused on domain transfer, on the IEMOCAP and MELD datasets.

**Additional Comments:**

None

**Audience:**

Yes

**Audience Explanation:**

While I believe that some individuals in TMLR’s audience would indeed be interested in the findings, I am afraid that this group is relatively small. The paper falls between two domains: emotion recognition and multimodal domain adaptation. The issue is that the paper is “too ML-oriented” for researchers working in emotion recognition (for instance, it contains very limited discussion of actual emotional phenomena), while at the same time being “too narrow” for general ML practitioners, as it is unclear which broader machine learning problems or themes could benefit from the proposed approach.

**Broader Impact Concerns:**

The paper addresses emotion recognition, which by nature can be invasive and potentially used unethically. However, there is nothing explicitly unethical in the paper itself, and the work is sufficiently ML-oriented that ethical concerns are not central to the contribution. That said, the paper does not include any discussion of ethical implications.

**Claims And Evidence:**

Yes

**Claims Explanation:**

The paper makes the following claims:

1. “First attempt to simultaneously mitigate domain shift and noisy label interference problems in MERC.” It is difficult to definitively confirm the “first” claim, but I have not found other works that clearly address both issues simultaneously.
2. “Improve the expressiveness of domain-invariant features.” The experimental results support this claim.
3. “Added a regularization constraint loss on top of the cross-entropy loss term to effectively suppress the model’s overlearning of noisy labels.” The reported results also support this claim. On the other hand, the actual amount of label noise generally cannot be accurately determined. In real-world settings, noise is not artificially added; rather, annotations are inherently noisy. Therefore, it remains debatable whether such a regularization term can effectively suppress real-world label noise without prior knowledge of the noise level and without negatively affecting correctly labeled samples.
4. The paper provides both theoretical analysis and experimental validation, and these appear to support the proposed approach

**Requested Changes:**

The main observation is that the paper is too long. However, once this is accepted, the paper is fair in the sense that it promises something and delivers it. The more difficult question is whether what it delivers is meaningful to a sufficiently broad audience.

---

### Review · Reviewer_bTQM · 2026-05-19

**Summary Of Contributions:**

This paper presents "Dual-branch Graph Domain Adaptation" (DGDA) for out-of-domain (cross-scenario) multi-modal emotion recognition. The approach combines hypergraph and path neural networks (HGNN/PathNN) to capture both local and global relationships. The system includes a domain adversarial discriminator to help generate more domain-invariant features, and a regularization loss to improve robustness to label noise. The approach is evaluated on cross-domain transfer between the IEMOCAP and MELD datasets, covering the joy, sadness, neutral and anger emotions. Results generally show improved weighted F1 scores compared to various baselines.

**Audience:**

Yes

**Audience Explanation:**

Yes. Emotion recognition, graph neural networks and out-of-domain generalization are of interest to some of the TMLR audience.

**Claims And Evidence:**

No

**Claims Explanation:**

Some additional baselines, in particular in-domain and noiseless results, would help interpret the data more clearly.

The authors claim that the weighted F1 metric "prevents both minority and majority classes from dominating the overall score", but this appears incorrect for frequent classes. Especially on MELD, the neutral class has a much higher impact on the overall results than sadness.

Results would be more convincing with higher performance on rare classes, although the baselines also generally struggle there too.

**Requested Changes:**

[Critical] Include in-domain performance

[Critical] Include 0% noise performance

[Critical] Update justification on the choice of metric

[Would strengthen] Also include macro-average F1

[Would strengthen] I would suggest moving the detailed proofs to the appendix given their length.

[Would strengthen] Tables 1-4 and 5-8 are nearly the same within each group (only changing the noise ratio) and take significant space. I would streamline the results. For example, within each group, you could keep 1 table for detailed per-class results, and another table for the overall impact of noise (with weighted and macro-average F1). Full results could be in the appendix.

[Would strengthen] In figure 5, reading some numbers in black on dark blue backgrounds is difficult. Consider improving the figure readability.

[Would strengthen] Mention technique (and hyper-parameters if any) used to generate 2D visualizations.

---

### Decision · Action_Editor_wZoG · 2026-06-24

**Recommendation:** Reject

**Audience:**

Yes

**Audience Explanation:**

The paper addresses the problem of multi-modal emotion recognition and out-of-domain generalization using graph neural networks and domain adaptation techniques. These topics are of interest to the TMLR audience.

**Claims And Evidence:**

No

**Claims Explanation:**

While some of the claims made in the paper have been supported by empirical and theoretical evidence, further validation is necessary to make the proposed method more convincing. The reviewers have mentioned that additional experiments (such as in-domain and noiseless performance) are necessary to appropriately understand the utility of the proposed method. Moreover, improved performance on rare classes will make the empirical results more convincing. Further, the performance needs to be evaluated in terms of the macroF1 or UAR metrics, together with standard deviation values. The organization and writing quality of the paper also need to improve.